# MICE-Bench: A Challenging and Comprehensive Benchmark for Multi-Reference Image Creation and Editing

**Siqi Luo** [1 2 *] **Huayu Zheng** [1 *] **Jianghan Shen** [2 3] **Yi Xin** [3 4] **Luxin Xu** [2 5] **Jiyao Liu** [2 6] **Xinyu Zhang** [2 5]
**Hang Zhou** [1] **Pengyu Xie** [2 3] **Xiaohui Li** [2 1] **Shuo Cao** [2 7] **Yuandong Pu** [2 1] **Junjun He** [2 4] **Bin Fu** [2 8] **Yihao Liu** [2]
**Yu Qiao** [2 4] **Guangtao Zhai** [1] **Yuewen Cao** [✉ 1] **Xiaohong Liu** [✉ 1 4]

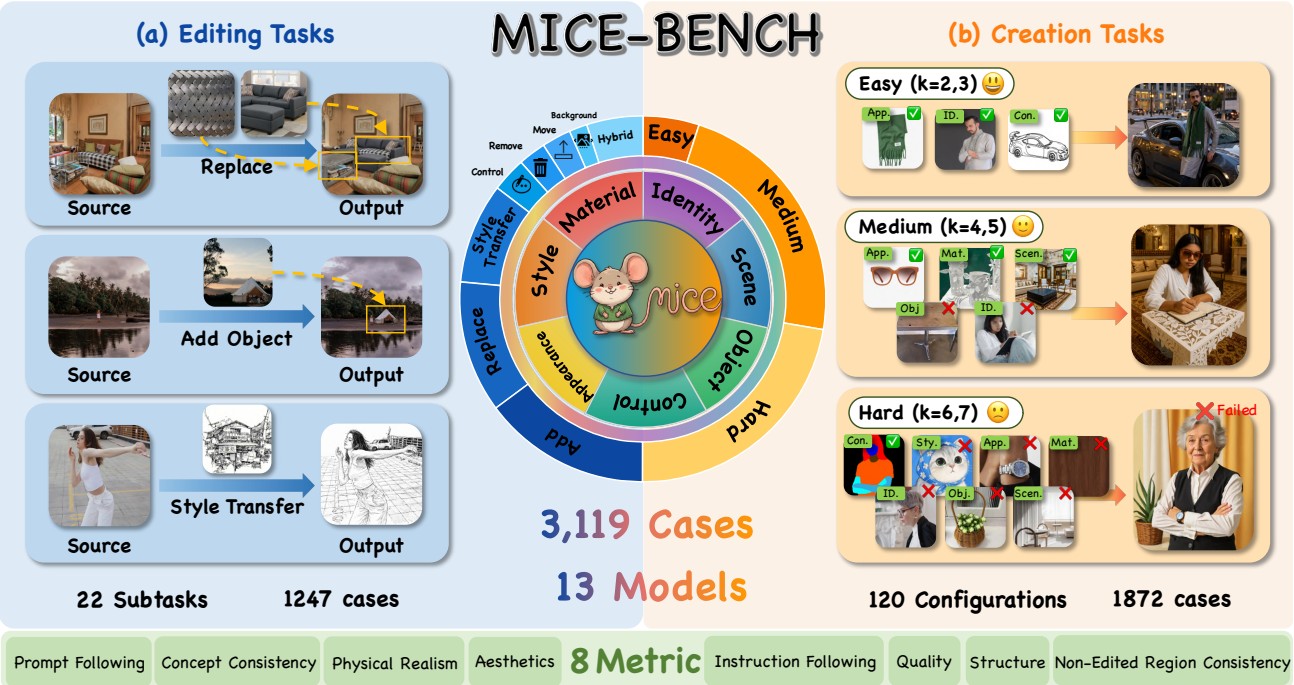

*Figure 1.* **Overview of our MICE-Bench, including evaluation tasks, data, and metrics.**

## Abstract

The paradigm of visual generation is rapidly shifting from single-image conditioning toward multi-image conditioning, making the ability to synthesize and edit images based on multiple visual references a critical capability. Despite this trend, existing benchmarks remain largely limited to single-reference scenarios or narrowly de-

fined tasks, leaving model behavior under complex multi-concept composition insufficiently explored. To bridge this gap, we introduce **MICE-Bench**, a comprehensive benchmark for **M**ulti-reference **I**mage **C**reation and **E**diting. The benchmark is designed around three core principles: 1) **heterogeneous concept composition** across seven visual dimensions; 2) varying levels of **constraint density**, ranging from dual-concept to seven-concept configurations; 3) **concept-centric data construction and benchmark evaluation**, enabling fine-grained analysis of interactions among multiple concepts. MICE-Bench consists of 3,119 high-quality test cases within a unified concept space. Using an 8-dimensional evaluation metric, we systematically evaluate 13 state-of-the-art models. Our results show that although closed-source models maintain a clear performance advantage, all models

*Equal contribution [1]Shanghai Jiao Tong University [2]Shanghai Artificial Intelligence Laboratory [3]Nanjing University [4]Shanghai Innovation Institute (SII) [5]University of Electronic Science and Technology of China [6]Fudan University [7]University of Science and Technology of China [8]Shenzhen Institutes of Advanced Technology, Chinese Academy of Sciences. Correspondence to: Yuewen Cao <caoyuewen@pjlab.org.cn>, Xiaohong Liu <xiaohongliu@sjtu.edu.cn>.

experience notable degradation in concept consistency and physical realism as concept complexity increases. This indicates that current models rely on superficial composition rather than genuine multi-concept synthesis, highlighting substantial room for future improvement. We will release all benchmark resources to support future research.

# 1. Introduction

The paradigm of visual generation is gradually shifting from single-image toward multi-image conditioning. Recent proprietary (OpenAI, 2025b; NanoBanana, 2025; ByteDance, 2025) and open-source models (Black Forest Labs, 2025; Wu et al., 2025a; AI et al., 2026; Xin et al., 2025a) have demonstrated emerging capabilities in extracting distinct concepts from multiple reference images for both creation and editing tasks. Such fine-grained control is critical for professional workflows like advertising and cinematography (Shuai et al., 2024; Wang et al., 2023), which require integrating heterogeneous visual dimensions such as identity, objects, style, and environmental context.

Despite the growing demand for multi-reference guidance, systematic characterization of model behavior under heterogeneous multi-reference constraints remains limited. Existing benchmarks exhibit three key limitations in their design. *(i) Task coverage and constraint density are insufficient*. Early benchmarks such as EditBench (Wang et al., 2023), and ICE-Bench (Pan et al., 2025) focus primarily on single-image editing tasks. Although more recent benchmarks such as OmniGen2 (Wu et al., 2025b) and DreamOmni2 (Xia et al., 2025) have begun to explore multi-reference tasks, the number of reference images is typically restricted to 2-4, and systematic evaluation across constraint density gradients is lacking. *(ii) Concept-level analysis is absent*. Existing benchmarks report results in aggregated form without differentiating performance across concept types, which limits fine-grained diagnosis of cross-concept interactions during synthesis. *(iii) Evaluation scale and coverage are limited*. Most existing benchmarks comprise only a few hundred test cases, making it difficult to sufficiently characterize model behavior in complex multi-concept composition settings.

To address the limitations above, we introduce **MICE-Bench**, a benchmark for **M**ulti-reference **I**mage **C**reation and **E**diting designed to enable fine-grained analysis of model behavior through controlled composition within a heterogeneous concept space spanning *Identity*, *Style*, *Material*, *Control*, *Scene*, *Object*, and *Appearance*. As illustrated in Fig. 1, MICE-Bench comprises two task families: 1) creation, which evaluates harmonious integration of multiple concepts, and 2) editing, which assesses modifications while ensuring preservation of unedited content.

We developed a fully automated pipeline to generate evaluation data for MICE-Bench. To ensure quality and mitigate potential biases from automated generation, we incorporated a rigorous human-in-the-loop verification process. Specifically, all generated components were subject to expert refinement and strict feasibility assessments to eliminate invalid samples. Ultimately, MICE-Bench comprises 3,119 high-quality evaluation cases.

To evaluate generative models on MICE-Bench, we also introduced a comprehensive evaluation framework. It includes 8 distinct dimensions that cover a wide spectrum of metrics, ranging from reference consistency and object realism to aesthetic quality. We conducted extensive experiments on 6 closed-source and 7 open-source models. Our results demonstrate that closed-source models maintain a significant lead in both multi-reference creation and editing tasks. This indicates that despite recent progress, there remains a substantial gap for open-source models to bridge.

Our main contributions are summarized as follows:

- **Systematic Benchmark Design.** We introduce MICE-Bench, a challenging benchmark designed to enable fine-grained analysis of multi-reference creation and editing across seven distinct visual concepts.
- **Rigorous Data Construction.** We construct a large-scale evaluation dataset comprising 3,119 cases using an automated pipeline with human-in-the-loop verification, ensuring high quality for complex constraints.
- **Comprehensive Evaluation Dimensions.** We design an 8-dimensional evaluation framework that assesses key aspects: concept consistency, semantic consistency, aesthetic, quality, and physical realism.
- **Extensive Evaluation & Insights.** We conduct a comprehensive evaluation of 13 leading models. Our results demonstrate that closed-source models currently maintain a distinct advantage, providing a clear roadmap for future open-source development.

# 2. Related Work

**Image Creation and Editing Models.** Image creation and editing aim to synthesize or modify images based on user-provided conditioning signals while preserving the integrity of irrelevant regions. A dominant paradigm involves subject-driven methods (Ruiz et al., 2022; Ye et al., 2023), which enable personalized image generation. On the editing side, instruction-based methods (Fu et al., 2024; Huang et al., 2024; Zhao et al., 2024; Xin et al., 2025b) have progressively enhanced editing precision and instruction adherence, while recent models (Labs et al., 2025;

*Table 1.* **Comparison with major benchmarks for reference-based image creation and editing.** Existing benchmarks lack systematic evaluation for multi-reference scenarios, support limited inputs, and ignore reference compatibility. Our benchmark expands the scale of references, tasks, and dataset size, while introducing improved metrics.

| Benchmark | Pub. | Size | Tasks | Refs | Gen. | Edit. | Metrics |
|---|---|---|---|---|---|---|---|
| EditBench (Wang et al., 2023) | CVPR 2023 | 240 | 1 | 1 | ✗ | ✓ | CLIP (Radford et al., 2021) |
| MagicBrush (Zhang et al., 2023) | NeurIPS 2023 | 1,053 | 9 | 1 | ✗ | ✓ | L1, L2, CLIP, DINO (Caron et al., 2021) |
| EmuEdit (Sheynin et al., 2024) | CVPR 2024 | 3,055 | 7 | 1 | ✗ | ✓ | L1, CLIP, DINO |
| I2EBench (Ma et al., 2024) | NeurIPS 2024 | 2,240 | 16 | 1 | ✗ | ✓ | GPT (OpenAI, 2023) |
| AnyEdit (Yu et al., 2025) | ICML 2025 | 1,250 | 25 | 1 | ✗ | ✓ | L1, CLIP, DINO |
| ICE-Bench (Pan et al., 2025) | ICCV 2025 | 6,538 | 31 | 1 | ✓ | ✓ | GPT (3 dim.) |
| ImgEdit-Bench (Ye et al., 2025) | NeurIPS 2025 | 811 | 14 | 1 | ✗ | ✓ | GPT (3 dim.), Fake Det. |
| OmniContext (Wu et al., 2025b) | Arxiv 2025 | 400 | 8 | 1∼3 | ✓ | ✗ | GPT 4.1 (3 dim.) |
| DreamOmni2 (Xia et al., 2025) | Arxiv 2025 | 319 | 16 | 2∼4 | ✓ | ✓ | Gemini 2.5, Doubao 1.6 |
| **MICE-Bench (Ours)** | **This paper** | **3,119** | **142** | **2∼7** | ✓ | ✓ | **Gemini 3 (5 dim.), UniPercept (3 dim.)** |

Liu et al., 2025) further improved complexity management through advanced multimodal understanding. However, these models have primarily focused on single-image editing. With the expansion of model capabilities, recent models (NanoBanana, 2025; OpenAI, 2025a; ByteDance, 2025; Black Forest Labs, 2025; Wu et al., 2025a) have begun to support multi-reference creation and editing through native multi-image inputs rather than compositional pipelines, enabling consistent and context-aware modifications across sets of related images. This advancement significantly enriches the user experience in applications.

**Image Creation and Editing Benchmarks.** The rapid advancement of image generation techniques has enabled reference-guided visual synthesis for both creation and editing, yet the evaluation landscape has remained predominantly editing-centric, with creation benchmarks being notably scarce. Early efforts (Zhang et al., 2023; Sheynin et al., 2024) relied on generic similarity metrics (e.g., CLIP and DINO), which often exhibit poor correlation with human perception, while subsequent benchmarks (Ye et al., 2025; Ma et al., 2024) improved evaluation standards through GPT-based protocols. More recently, benchmarks such as OmniContext (Wu et al., 2025b) and DreamOmni2 (Xia et al., 2025) aim to address multi-image creation and editing, yet they typically involve few reference images (2–4), cover a narrow range of tasks, and lack extensive model validation. To address these limitations, we introduce MICE-Bench, featuring diverse task categories spanning both creation and editing, flexible reference scales, and rigorous validation across numerous models and metrics.

## 3. MICE-Bench

To systematically evaluate multi-reference creation and editing under diverse heterogeneous concept composition, we construct **MICE-Bench** based on three core design principles (§3.1). The benchmark comprises two task families: *creation* and *editing*, enabling characterization of model capabilities under distinct constraint environments (§3.2). To realize this, we implement a concept-centric data construction pipeline combining AI planning with expert validation (§3.3), alongside multi-dimensional metrics for comprehensive analysis (§3.4).

### 3.1. Design Principles

MICE-Bench is designed to systematically characterize model capabilities under multi-reference image settings, covering diverse concept composition configurations. Guided by this objective, we organize our benchmark design around three core principles.

**Heterogeneous Concept Composition.** We define a concept space $\mathcal{C} = \{$*Identity*, *Object*, *Style*, *Scene*, *Material*, *Control*, *Appearance*$\}$ (details in Appendix A.1), where each reference image corresponds to a distinct concept. We leverage this taxonomy to benchmark model performance across creation and editing tasks. This design supports fine-grained characterization of model behavior, revealing how distinct concepts interact under joint constraints.

**Constraint Density of Concepts.** To analyze how model behavior evolves as concept complexity increases, we organize tasks along *constraint density of concepts*, spanning from $K = 2$ to $K = 7$ concepts. This stratified design exposes systematic trends in model performance under growing compositional pressure.

**Concept-Centric Construction and Evaluation.** We design the data construction pipeline in a *concept-centric* manner, where each reference is explicitly bound to its underlying concept types and composition structure. By integrating concept-driven automated construction with expert verification, and subsequently mapping model outputs to concept-level metrics, the benchmark ensures rigorous

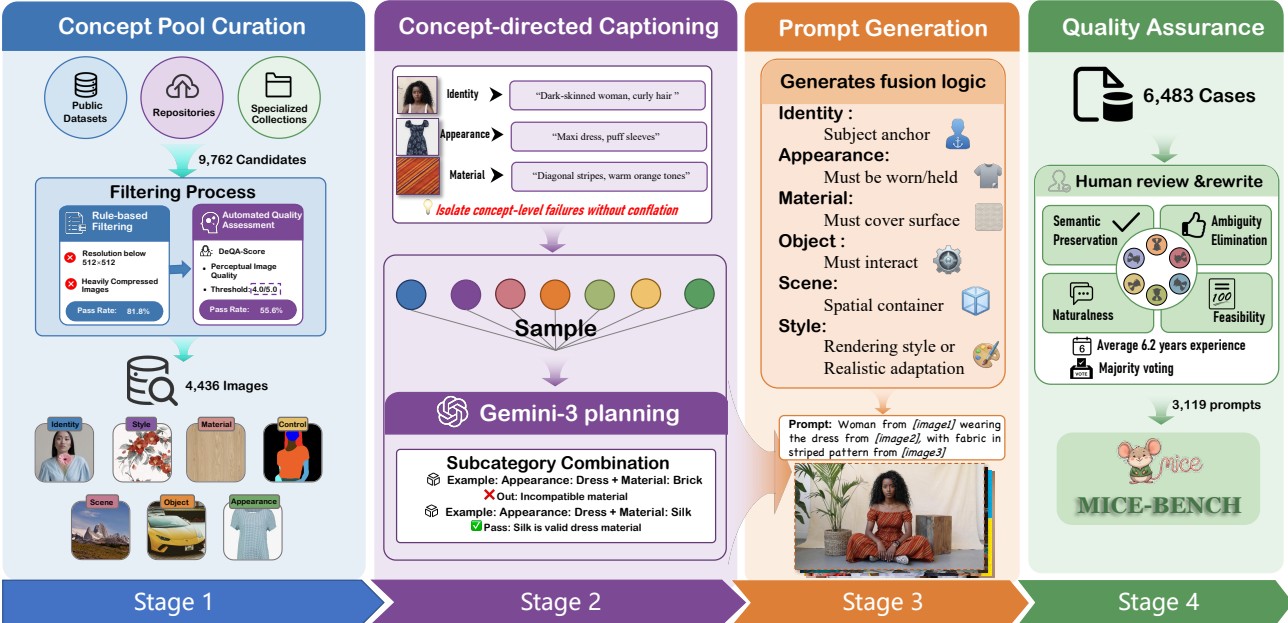

*Figure 2.* **Data construction pipeline for MICE-Bench.** We employ a four-stage process: concept pool curation with dual-stage validation, concept-directed captioning for in-depth diagnosis, prompt generation, and expert quality assurance.

quality control while maintaining natural scalability.

### 3.2. Task Design

**Image Creation Task.** This task evaluates the ability to synthesize novel images by composing heterogeneous visual concepts from multiple reference images. Given textual prompt $p$ and reference image set $R = \{I_1, \ldots, I_K\}$, the model ***generates a fused image that harmoniously integrates all specified concepts*** $\{c_j\}_{j=1}^K \in \{I_j\}_{j=1}^K$ while maintaining visual coherence and physical realism.

Creation tasks require models to resolve concept competition without predefined spatial layouts. This challenge intensifies as density $K$ increases, often leading to dominant concepts suppressing fragile ones. We construct ***120 task configurations*** through stratified sampling by enumerating concept-type compositions across the seven concept dimensions, i.e., $\sum_{K=2}^7 \binom{7}{K} = 120$, spanning constraint densities with $K \in \{2, 3, 4, 5, 6, 7\}$ and yielding 1,872 test cases. As shown in Figure 1, this design operationalizes the constraint density of concepts. Lower densities ($K = 2, 3$) correspond to easy settings with basic compositional complexity. Intermediate densities ($K = 4, 5$) represent medium difficulty, introducing increased interactions where interference commonly emerges. High densities ($K = 6, 7$) constitute hard settings, simulating extreme heterogeneous composition pressures.

**Image Editing Task.** This task evaluates a model's ability to modify a given target image according to textual in-

structions and visual concepts provided by reference images. Given a target image $I_{target}$, a textual prompt $p$, and a reference set $R = \{I_1, \ldots, I_n\}$, where each reference $I_j$ provides a visual concept $c_j$ that may belong to the same or different concept types, the model ***generates a modified target image that simultaneously satisfies the following requirements***: (1) executing the instruction $p$; (2) incorporating the concept constraints specified by the reference set $R$; and (3) preserving unmodified content from the target image $I_{target}$.

Unlike creation tasks, editing requires precise spatial control: models must localize modifications while preserving lighting, viewpoint, and content consistency in non-edited regions. Editing operations span both localized modifications (e.g., *object replacement, addition*) and global transformations (e.g., *style transfer, background changes*). We design ***22 atomic editing subtasks*** (shown in Figure 3 (b), with Hybrid treated as a single subtask) organized hierarchically into eight operation categories: ***four local editing operations*** (*Add*, *Replace*, *Move*, *Remove*) and ***four global editing operations*** (*Style*, *Background*, *Controllable*, *Hybrid*). All editing subtasks operate at concept density $K \leq 3$ with $n \in [1, 4]$ reference images, yielding 1,247 test cases.

### 3.3. Dataset Construction

**Concept Pool Curation.** We collected 9,762 real images from public datasets, creative commons platforms (ensuring legal license), and lifestyle photography photographs

taken by our team (details in Appendix A.2). The filtering process involves two stages: *1) Rule-based Filtering*, enforcing a minimum resolution of $512\times512$ and a bit rate above 0.15; and *2) Automated Quality Assessment* via DeQA-Score (You et al., 2025) with a threshold of 4.0/5.0. Ultimately, 4,436 pass validation, yielding a 45.4% acceptance rate. To ensure accuracy, each validated image was manually annotated according to our concept taxonomy.

**Concept-Directed Captioning.** Standard image captions often conflate multiple attributes (e.g., "*a woman in red dress in forest*" entangles *Identity, Appearance, and Scene*). However, a reference image is typically intended to convey a single specific concept. To address this, we introduce concept-directed captioning via Gemini-3, which constrains ***descriptions to only the target concept***. For instance, Identity captions specify *facial geometry*, excluding clothing or background, whereas Material captions describe *texture*, ignoring object shape. This decoupling allows us to evaluate individual concepts independently.

**Concept Automated Planning & Creation / Editing Prompt Generation.** To construct complex multi-reference tasks, we developed an automated pipeline powered by Gemini-3. Initially, we randomly sample 2-7 distinct instances from the concept pool as reference inputs. Leveraging the previously generated concept-directed captions, Gemini-3 acts as a logical planner to synthesize these inputs. It establishes a ***fusion logic*** by assigning specific semantic roles to each reference—for instance, designating an *Identity* image as the subject anchor, a *Scene* image as the background context, or a *Material* image to define a surface texture. Following this planning, Gemini-3 generates coherent prompts that explicitly instruct the integration of these attributes (e.g., *a woman from [image 1] wearing the dress from [image 2] with the texture from [image 3]*), covering both creation and editing scenarios.

A notable exception is the *Removal* editing task, where random sampling is unsuitable because the reference must already exist within the target image. To address this, we employ an ***inclusion-based strategy***: we first select a target image and then retrieve one of its existing annotated concepts to serve as the reference.

**Quality Assurance.** All <Reference Images, Creation / Editing Prompt> pairs undergo comprehensive review and rewriting by six professional annotators. This process adheres to three key principles: *concept combination rationality*, *prompt correctness*, and *feasibility*. To facilitate this, we developed an annotation system, as detailed in Appendix A.3. Furthermore, each data instance undergoes cross-validation by three experts to ensure reliability.

**Numerical Statistics.** Our MICE-Bench consists of 1,872 instances and 1247 instances for creation and editing respectively. Figure 3 summarizes the detailed distribution of

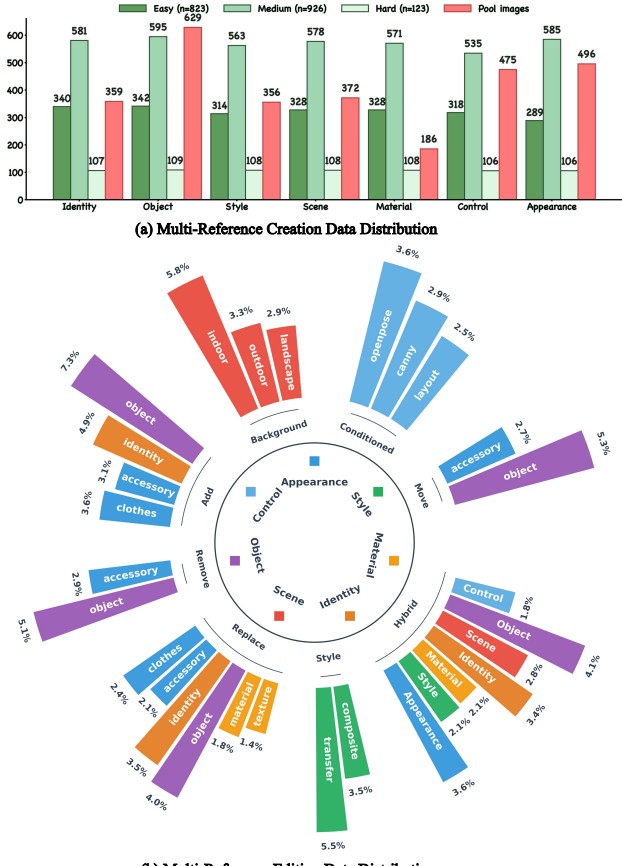

(a) Multi-Reference Creation Data Distribution

(b) Multi-Reference Editing Data Distribution

*Figure 3.* **MICE-Bench Data Distribution Statistics.**

MICE-Bench. For the creation task, samples are balanced across seven core concepts and three difficulty levels. *Pool images* in Figure 3(a) denotes the number of curated reference images *sampled and used* for the creation task in each concept category.

## 3.4. Evaluation Metrics

To assess multi-reference creation and editing performance, we established 8 evaluation dimensions: *prompt (for creation) / instruction (for editing) following, concept consistency, physical realism, image aesthetics / quality / structure, and non-edited region consistency.*

**Prompt Following.** Tailored for creation tasks, this metric assesses alignment between generated images and prompts containing visual placeholders (e.g., "a dog in [reference image]. . ."). As standard scorers cannot interpret these placeholders, we convert them into text description (e.g., "a long-eared spotted dog. . .") and then calculate semantic similarity on the <Rewritten Text, Generated Image> using Gemini-3 (more details in Appendix B.1).

**Instruction Following.** For editing tasks, we evaluate the accuracy of instruction execution on the target image. Distinct from the above creation metric, this assessment oper-

*Table 2.* **Quantitative comparison results on multi-reference creation tasks.** ■ ■ indicate the best and second results, respectively.

| Model | Prompt Following | Concept Consistency | Physical Realism | Image Aesthetics | Image Quality | Image Structure | Overall |
|---|---|---|---|---|---|---|---|
| *Closed-Source Models* | | | | | | | |
| Seedream 4.0 | 8.67 | 8.20 | 7.45 | 6.03 | 6.88 | 4.88 | 7.02 |
| Seedream 4.5 | 8.95 | 8.70 | 8.11 | **6.82** | **7.37** | 4.86 | 7.47 |
| NanoBanana | 8.54 | 8.14 | 7.71 | 6.61 | 7.22 | 4.94 | 7.19 |
| NanoBanana 2 | 9.01 | 8.57 | 8.49 | 6.58 | 7.22 | 4.94 | 7.48 |
| GPT-Image 1.0 | 8.87 | 8.62 | 8.59 | 6.19 | 7.05 | 4.87 | 7.37 |
| GPT-Image 1.5 | **9.07** | **8.81** | **8.64** | **6.82** | 7.32 | 4.88 | **7.59** |
| *Open-Source Models* | | | | | | | |
| BAGEL | 7.49 | 5.80 | 7.45 | 5.64 | 6.48 | 4.65 | 6.25 |
| OmniGen2 | 6.61 | 5.85 | 6.90 | 5.36 | 6.41 | 4.34 | 5.91 |
| DreamOmni2 | 6.00 | 4.75 | 7.88 | 5.72 | 6.65 | 4.34 | 5.89 |
| FLUX.2 [dev] | 8.78 | 7.65 | 8.35 | 6.13 | 7.00 | 4.69 | 7.10 |
| FLUX.2 [klein] | 8.74 | 7.69 | 8.20 | 6.72 | 7.25 | **4.96** | 7.26 |
| Qwen-Edit-2509 | 5.34 | 5.62 | 6.21 | 5.41 | 5.88 | 4.10 | 5.43 |
| Qwen-Edit-2511 | 6.88 | 6.55 | 7.44 | 6.22 | 7.11 | 4.78 | 6.50 |

*Table 3.* **Quantitative comparison on multi-reference editing (local) tasks.** ■ ■ indicate the best and second results, respectively.

| Model | Add IF ↑ | Add CC ↑ | Add NERC ↑ | Add PR ↑ | Move IF ↑ | Move CC ↑ | Move NERC ↑ | Move PR ↑ | Remove IF ↑ | Remove CC ↑ | Remove NERC ↑ | Remove PR ↑ | Replace IF ↑ | Replace CC ↑ | Replace NERC ↑ | Replace PR ↑ | Overall (Local) |
|---|---|---|---|---|---|---|---|---|---|---|---|---|---|---|---|---|---|
| *Closed-Source Models* | | | | | | | | | | | | | | | | | |
| Seedream 4.0 | 8.06 | 8.79 | 6.05 | 7.17 | 8.84 | 8.64 | 6.41 | 9.15 | 8.21 | 4.65 | 7.42 | 8.61 | 8.10 | 7.87 | 6.53 | 7.65 | 7.63 |
| Seedream 4.5 | 8.44 | 9.20 | 5.80 | 8.04 | 9.26 | 9.13 | 5.94 | 9.32 | **8.55** | 5.33 | 7.81 | 9.11 | 8.33 | 8.78 | 6.43 | 8.41 | 7.99 |
| NanoBanana | 8.05 | 8.49 | 5.29 | 7.67 | 8.35 | 7.53 | 5.62 | 9.28 | 7.58 | 5.64 | 6.84 | 9.19 | 7.50 | 7.37 | 4.07 | 8.30 | 7.30 |
| NanoBanana 2 | 9.07 | **9.60** | **7.45** | 8.01 | 8.79 | **9.63** | 8.69 | 8.89 | 8.48 | 5.97 | **9.09** | 9.21 | **9.10** | **9.34** | 7.53 | 7.94 | **8.55** |
| GPT-Image 1.0 | 8.28 | 9.11 | 4.06 | 8.50 | 8.56 | 7.95 | 3.21 | 9.22 | 6.92 | 5.30 | 3.44 | **9.39** | 8.58 | 9.26 | 4.84 | 8.76 | 7.21 |
| GPT-Image 1.5 | **9.16** | 9.45 | 7.33 | **8.86** | 8.71 | 8.58 | 6.76 | **9.35** | 8.53 | 4.43 | 8.19 | 9.12 | 9.07 | 9.26 | 7.29 | **8.85** | 8.31 |
| *Open-Source Models* | | | | | | | | | | | | | | | | | |
| BAGEL | 4.32 | 5.08 | 1.39 | 6.43 | 6.85 | 6.19 | 2.74 | 7.78 | 3.59 | 3.71 | 1.16 | 7.09 | 3.20 | 3.39 | 0.71 | 6.60 | 4.39 |
| OmniGen2 | 5.78 | 5.73 | 1.36 | 7.08 | 6.57 | 5.93 | 1.03 | 8.36 | 3.78 | 6.89 | 0.62 | 8.63 | 5.42 | 4.46 | 1.07 | 7.59 | 5.02 |
| DreamOmni2 | 6.39 | 6.23 | 7.28 | 7.40 | 7.15 | 6.81 | 7.91 | 8.71 | 6.03 | 5.60 | 6.53 | 9.06 | 6.70 | 4.88 | 6.91 | 8.48 | 7.00 |
| FLUX.2 [dev] | 8.33 | 8.59 | 7.37 | 7.78 | 8.41 | 8.90 | 7.74 | 8.79 | 7.79 | 5.06 | 7.66 | 8.96 | 8.41 | 8.36 | **7.59** | 8.19 | 8.00 |
| FLUX.2 [klein] | 7.91 | 8.46 | 6.99 | 8.01 | 8.22 | 8.50 | **8.79** | 9.12 | 6.24 | **7.52** | 7.62 | 9.03 | 7.91 | 7.86 | 6.63 | 8.39 | 7.95 |
| Qwen-Edit-2509 | 6.14 | 7.11 | 3.90 | 7.03 | 6.42 | 6.01 | 5.82 | 7.21 | 4.46 | 5.25 | 3.63 | 7.98 | 4.95 | 4.75 | 2.95 | 7.07 | 5.67 |
| Qwen-Edit-2511 | 7.13 | 7.66 | 4.75 | 7.35 | 8.08 | 8.18 | 8.35 | 8.91 | 5.23 | **7.43** | 7.50 | 8.84 | 6.68 | 6.42 | 4.39 | 7.67 | 7.16 |

ates on a triplet: <Target Image, Editing Instruction, Edited Image>. Consistent with the above metric, visual placeholders within instructions are also converted into textual descriptions prior to evaluation via Gemini-3.

**Concept Consistency.** Preserving the distinct visual identity of each reference is critical in multi-reference creation and editing. This metric evaluates the fidelity with which specific concepts (e.g., objects, styles) are retained in the output. The assessment operates on the <Reference Images, Generated Image>, employing Gemini-3 to quantify visual similarity and measure the preservation of defining characteristics (more details in Appendix B.2).

**Physical Realism.** This metric assesses whether the generated images adhere to real-world physical laws, such as spatial incoherence, human deformities, etc. Evaluation is performed directly on the generated image, with Gemini 3 assigning a rigorous scoring criteria in Appendix B.3.

**Image Aesthetics, Quality, and Structure.** Image aesthetics and quality are fundamental metrics in image generation. To assess these, we employ UniPercept (Cao et al., 2025), a state-of-the-art MLLM evaluator. Distinct from traditional single-metric evaluator, UniPercept simultaneously quantifies aesthetics and quality. Furthermore, it can also assess image structure, a metric evaluating the rationality of composition and layout. We incorporate all three dimensions for a comprehensive visual assessment.

**Non-Edited Region Consistency.** This metric verifies that regions not targeted by the editing instruction (e.g., backgrounds or unrelated objects) remain unaltered. Consistent with the instruction following metric, we utilize the

*Table 4.* **Quantitative comparison on multi-reference editing (global) tasks.** ■ ■ indicate the best and second results, respectively.

| Model | Background | | | | Style | | | | Controllable | | | | Hybrid | | | | Overall |
|---|---|---|---|---|---|---|---|---|---|---|---|---|---|---|---|---|---|
| | IF ↑ | CC ↑ | NERC ↑ | PR ↑ | IF ↑ | CC ↑ | NERC ↑ | PR ↑ | IF ↑ | CC ↑ | NERC ↑ | PR ↑ | IF ↑ | CC ↑ | NERC ↑ | PR ↑ | (Global) |
| *Closed-Source Models* | | | | | | | | | | | | | | | | | |
| Seedream 4.0 | 6.42 | 9.14 | 5.38 | 7.00 | 7.84 | 9.29 | 5.57 | 6.57 | 7.46 | 7.27 | 7.84 | 7.64 | 7.58 | 7.95 | 6.32 | 7.37 | 7.29 |
| Seedream 4.5 | 6.55 | 9.23 | 8.75 | 8.18 | 8.40 | 9.46 | 7.26 | 7.00 | 7.75 | 8.01 | 6.87 | 8.55 | 7.90 | 8.48 | 6.88 | 8.10 | 7.96 |
| NanoBanana | 8.28 | 9.38 | 7.58 | 5.63 | 7.88 | 8.75 | 6.11 | 7.43 | 7.55 | 6.59 | 5.38 | 9.09 | 7.78 | 7.55 | 5.58 | 7.80 | 7.40 |
| NanoBanana 2 | 8.87 | 9.47 | 7.25 | 6.38 | 8.12 | 9.38 | 8.37 | 6.62 | 8.81 | 7.84 | 8.57 | 9.27 | 8.65 | 8.64 | 8.02 | 7.74 | 8.25 |
| GPT-Image 1.0 | 8.50 | 9.29 | 7.08 | 7.43 | 8.73 | 9.55 | 4.90 | 8.18 | 7.91 | 7.43 | 6.86 | 9.20 | 8.18 | 8.27 | 5.09 | 8.50 | 7.82 |
| GPT-Image 1.5 | 8.98 | 9.40 | 7.75 | 8.10 | 8.90 | 9.51 | 7.30 | 8.24 | 8.88 | 8.23 | 8.05 | 9.25 | 8.93 | 8.47 | 7.48 | 8.66 | 8.50 |
| *Open-Source Models* | | | | | | | | | | | | | | | | | |
| BAGEL | 2.25 | 5.00 | 0.00 | 7.02 | 5.39 | 5.43 | 1.43 | 7.10 | 3.24 | 3.82 | 1.45 | 6.40 | 3.71 | 4.39 | 1.08 | 6.97 | 4.04 |
| OmniGen2 | 7.47 | 8.97 | 5.17 | 6.85 | 6.29 | 6.05 | 2.33 | 6.30 | 5.52 | 5.40 | 1.93 | 7.21 | 5.76 | 5.86 | 1.86 | 7.11 | 5.63 |
| DreamOmni2 | 8.53 | 8.03 | 5.58 | 7.83 | 5.15 | 5.65 | 7.11 | 6.71 | 6.40 | 4.97 | 7.29 | 8.56 | 6.36 | 5.69 | 6.81 | 7.88 | 6.78 |
| FLUX.2 [dev] | 8.00 | 9.22 | 8.83 | 5.83 | 7.61 | 9.22 | 8.44 | 6.16 | 7.04 | 7.40 | 6.69 | 8.00 | 7.69 | 8.11 | 7.57 | 7.25 | 7.69 |
| FLUX.2 [klein] | 6.55 | 9.02 | 4.75 | 7.52 | 8.04 | 8.94 | 6.02 | 7.87 | 7.18 | 6.59 | 6.51 | 7.95 | 7.32 | 7.84 | 6.30 | 8.04 | 7.28 |
| Qwen-Edit-2509 | 8.22 | 9.11 | 7.33 | 6.12 | 6.63 | 6.23 | 6.54 | 6.63 | 3.94 | 4.90 | 2.64 | 7.37 | 5.43 | 5.86 | 4.28 | 6.89 | 6.13 |
| Qwen-Edit-2511 | 7.63 | 9.17 | 5.17 | 7.67 | 7.26 | 7.98 | 7.43 | 7.14 | 5.39 | 6.13 | 3.85 | 6.82 | 6.48 | 7.20 | 5.30 | 7.40 | 6.75 |

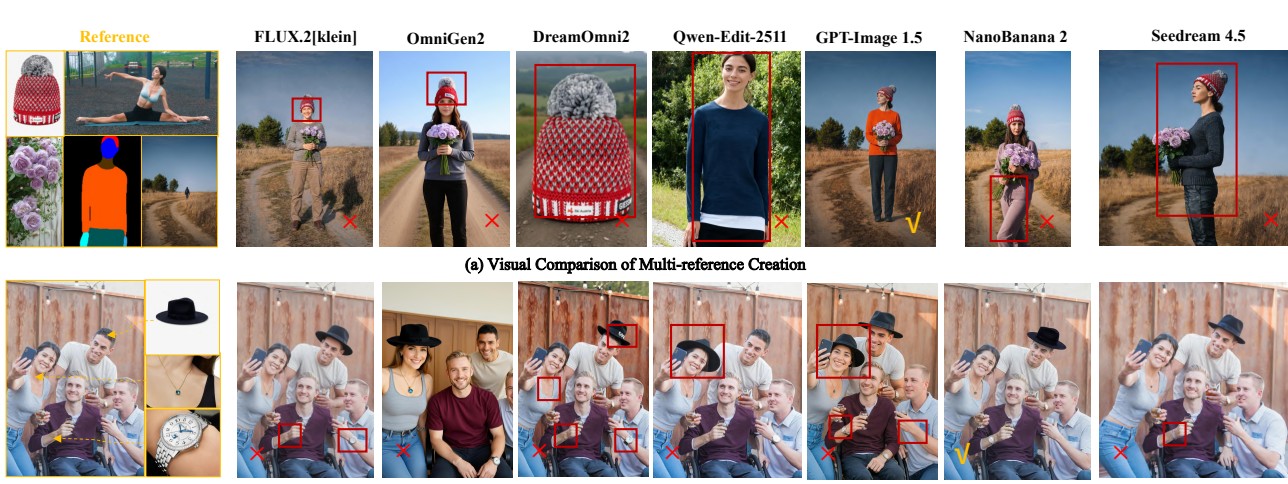

(a) Visual Comparison of Multi-reference Creation

(b) Visual Comparison of Multi-reference Editing

*Figure 4.* **Examples of creation and editing results on 7 models.**

<Target Image, Editing Instruction, Edited Image> triplet, but specifically focus the evaluation on the consistency of non-edited regions (see Appendix B.4 for scoring criteria).

## 4. Experiment

### 4.1. Evaluation Settings

**Evaluation Models.** We evaluate 13 representative models capable of processing multi-reference inputs, divided into two categories: 1) 7 open-source models (BAGEL (Deng et al., 2025), OmniGen2 (Wu et al., 2025b), DreamOmni2 (Xia et al., 2025), FLUX.2 [dev/klein] (Black Forest Labs, 2025), Qwen-Edit [2509/2511]) (Wu et al., 2025a)) and 2) 6 closed-source models (Seedream [4.0/4.5] (ByteDance, 2025), NanoBanana [Standard/Pro] (NanoBanana, 2025), GPT-Image [1.0/1.5] (OpenAI, 2025b)). The label *NanoBanana 2* in figures and tables corresponds to NanoBanana

Pro. Notably, we incorporate iterative versions within specific model families to assess evolutionary improvements in multi-reference creation and editing. Details hyperparameters for all models are in Appendix D.1.

### 4.2. Evaluation Results

**Multi-Reference Creation Results.** Quantitative results are presented in Table 2. Generally, *closed-source models maintain a performance lead, though top-tier open-source models demonstrate competitive capabilities*. GPT-Image 1.5 achieves the highest overall score (7.59), exhibiting comprehensive dominance by securing top ranks in Semantic Alignment (Prompt Following: 9.07 and Concept Consistency: 8.81) and Physical Realism (8.64). NanoBanana 2 follows as a strong runner-up (Overall: 7.48). Notably, Seedream 4.5 distinguishes itself in visual presentation, achieving the highest score in Image Quality (7.37) and tying for the best performance in Image Aes-

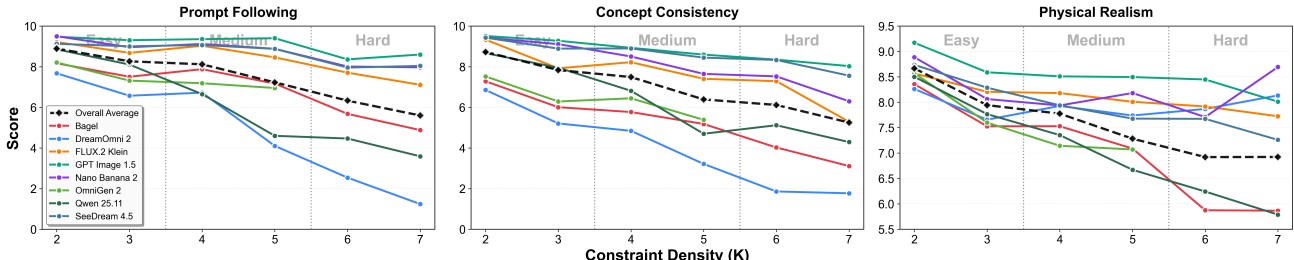

*Figure 5.* **A Comprehensive Comparison of Open Source Models and Closed Source Models.**

*Figure 6.* **Trends in Model Performance Changes with Increasing Concept Complexity.**

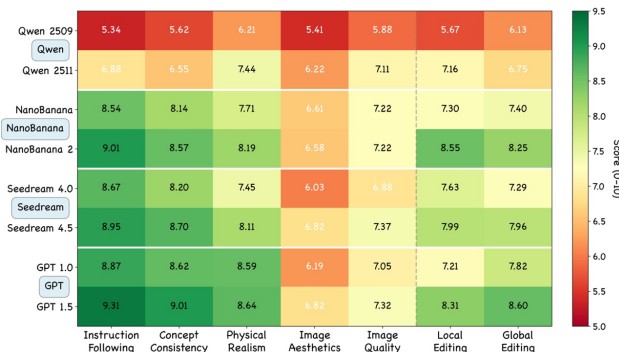

*Figure 7.* **Performance Comparison of Model Evolution.**

thetics (6.82). Notably, while most open-source models lag behind closed-source models, the FLUX.2 series emerges as a strong exception, outperforming many closed-source models. ***Regarding model evolution, we observe consistent performance gains across families*** such as Seedream, NanoBanana, and GPT-Image, with Qwen-Edit demonstrating the most significant improvements (see Section 4.3 for more detailed analysis). We further provide concept-level analysis and degradation trends in Appendix C.

**Multi-Reference Editing Results.** Quantitative results for multi-reference editing are presented in Table 3 (Local) and Table 4 (Global). In Local Editing, NanoBanana 2 achieves the state-of-the-art overall score (8.55), demonstrating superior precision in local operations. Conversely, the leaderboard shifts in Global Editing, where GPT-Image 1.5 takes the lead (8.50), dominating in global tasks such as style transfer. Notably, the open-source FLUX.2 se-

ries remains highly competitive, effectively narrowing the gap with closed-source models. In addition, the cross-task analysis analysis further reveals that ***strong generative priors do not guarantee editing capability***. While models like BAGEL demonstrate competence in creation, their performance collapses in editing tasks. Most notably, BAGEL drops to a zero score in Background NERC score. This suggests that without specific mechanisms for structural retention, such models treat editing merely as "re-generation," disregarding source image-reference constraints. In contrast, top-tier models effectively disentangle generation from preservation, validating the necessity of evaluating creation and editing tasks separately.

**Qualitative Results.** Figure 4 presents the qualitative results of multi-reference generation. As highlighted by the red boxes, most models struggle to effectively integrate diverse inputs, often resulting in missing attributes or distorted details. These limitations indicate that current methods still fall short of practical application standards, leaving significant room for improvement.

### 4.3. Analysis and Discussion

**How Far Are Open-Source Models Behind Closed-Source Models?** Figure 5 visualizes the performance disparity. Closed-source models establish a dominant upper bound across all dimensions, forming a comprehensive "performance envelope," with the gap being most pronounced in tasks requiring precise control. Specifically, while top-tier open-source models (e.g., Flux.2, Qwen-Edit) have narrowed the gap in basic visual *Quality* and

*Aesthetics*, they exhibit significant degradation in complex editing tasks such as *Move*, *Remove*, and *Hybrid*. This indicates that although the open-source community has mastered high-fidelity image synthesis, matching the rigorous logical adherence and fine-grained controllability of proprietary systems remains a substantial challenge.

**Does Model Evolution Enhance Multi-Reference Capabilities?** We analyze the performance trajectory across several model families (e.g., Qwen, NanoBanana, Seedream, GPT-Image), as shown in Figure 7. The heatmap reveals a distinct positive trend: newer iterations consistently outperform their predecessors across nearly all metrics. This progression is visually evident as the color spectrum shifts from lower-scoring regions (red/orange) to higher-scoring zones (green) in updated versions. This confirms that recent architectural and training advancements are effectively addressing key bottlenecks in multi-condition generation.

**Performance Degradation with More Concepts.** We investigate the correlation between the number of reference concepts ($K \in [2, 7]$) and model performance. As illustrated in Figure 6, there is a clear inverse correlation: ***as the number of concepts increases, performance across most metrics exhibits a monotonic decline***. These results indicate that integrating a high number of concepts poses a significant challenge to the models. Specifically, as the number of references scales up to the "Hard" level, models increasingly struggle to preserve the fidelity of individual concepts while maintaining consistency, exposing a capacity limitation in current generative models. More experimental results are provided in Appendix D.4.

## 5. Conclusion

In this work, we introduce MICE-Bench, a comprehensive benchmark tailored for multi-reference image creation and editing. To ensure both high data quality and logical coherence, we employ a rigorous four-stage construction pipeline. Our extensive evaluation of SOTA models reveals that current methods still face significant challenges in handling complex concept combinations while preserving physical realism. We believe MICE-Bench provides a reliable standard for the community, fostering future advancements multi-modal image generation. Limitations and future work are provided in Appendix D.6.

## Acknowledgements

The work was supported by the National Natural Science Foundation of China under Grant 62572317. This work was also supported by Shanghai Artificial Intelligence Laboratory.

## Impact Statement

This benchmark facilitates the development of precise multi-reference generation tools for creative applications. However, high-fidelity identity and style transfer capabilities carry risks of misuse, such as deepfakes. We urge researchers to utilize MICE-Bench to enhance model safety and reliability alongside performance.

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

# MICE-Bench: A Challenging and Comprehensive Benchmark for Multi-Reference Image Creation and Editing

## Appendix

## A. More Details on Benchmark Dataset Construction

### A.1. Detailed Concept Description

We define a concept space with 7 categories. Each concept details and fine-grained subcategories are listed below:

- **Identity**: Group (family, team, crowd), Individual (close-up, full-body portrait).

- **Object**: Animal (mammals, birds, aquatic life), Architecture (residential, commercial, bridges), Furniture (seating, tables, storage), Plant (flowers, trees, shrubs), Vehicle (cars, bicycles, aircraft), Text (signage, overlays).

- **Scene**: Urban (cityscapes, street views), Indoor (living room, office, workspace), Nature (landscapes, forests), Complex Structures (multi-room layouts).

- **Appearance**: Accessory (hats, glasses, jewelry, scarves), Clothing (casual, formal, sportswear, uniforms), Makeup (natural, artistic, theatrical).

- **Control**: Background Canny (edge maps), Viewpoint (orientation, camera angle), Foreground Canny, Lighting (natural, artificial, cinematic), Human Pose (skeletal keypoints, gestures).

- **Style**: Artistic Style (impressionism, surrealism, abstract), Background Style (scenic rendering), Object Style (stylized textures, caricatures).

- **Material**: Material (wood, metal, glass, fabric), Texture (rough, polished, knitted, grained).

### A.2. Concept Pool Construction Details

The construction of the concept pool involves two stages: data collection and data filter.

**Data Collection Sources.** We collect images from multiple curated sources to ensure both diversity and quality: (1) **Public datasets**, including CelebA (Liu et al., 2015) and FFHQ (Karras et al., 2019) for Identity concepts, WikiArt (Tan et al., 2018) for Style, and DTD (Cimpoi et al., 2014) for Material concepts; (2) **Creative commons platforms**, such as Flickr [1], Unsplash [2], and Pexels [3], with legal licensing; (3) **Lifestyle photography by our team**. We curate realistic scenarios to better align with practical user applications.

**Data Filter Pipeline.** We employ a two-stage process to ensure data integrity:

(1) **Automated rule-based filtering** discards images with low resolution ($< 512 \times 512$), visible watermarks, or significant compression artifacts. In addition, we removed images with bit information rate of less than 0.15, as these images were mostly heavily compressed. The specific calculations are as follows:

$$I_{bit} = \frac{Image\ File\ Size}{Height \times Width} < 0.15 \tag{1}$$

(2) **Automated quality scoring** uses DeQA-Score (You et al., 2025) to assess image quality, retaining only samples with a score $\geq 4.0/5$. DeQA-Score is the state-of-the-art (SOTA) model, which can predict the score distribution that closely aligns with human annotations.

Out of 9,762 initial candidates, 4,436 images passed all stages (45.4% pass rate). The primary reasons for exclusion were insufficient resolution (7.3%), low quality scores (47.3%). This rigorous pipeline ensures the pool provides high-quality, distinct references for evaluating multi-concept generation.

---

[1] https://www.flickr.com/
[2] https://unsplash.com/
[3] https://www.pexels.com/

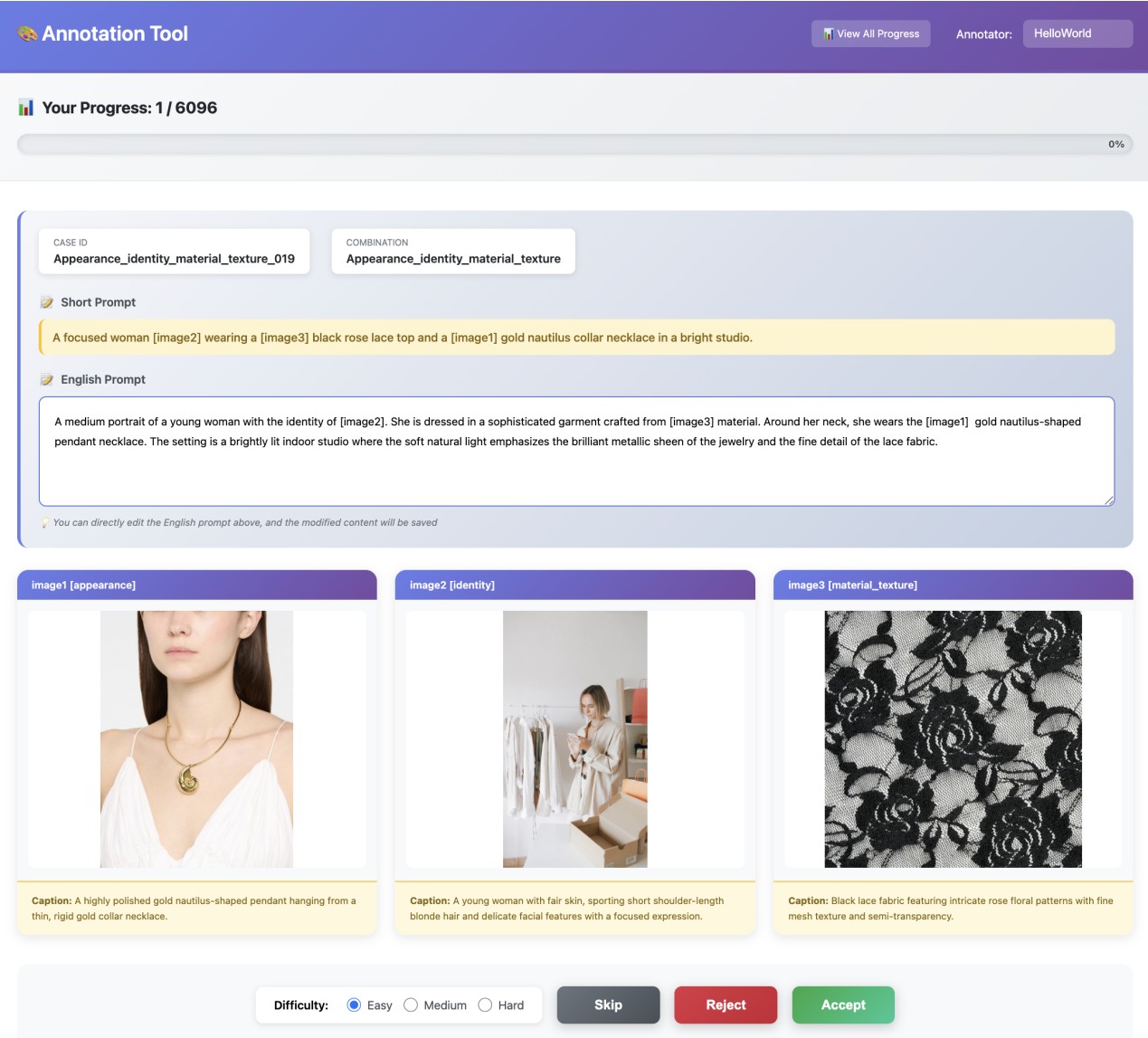

*Figure 8.* **GUI interface of human annotation system.**

## A.3. Human Annotation System

Figure 8 shows the GUI interface of the human annotation system designed to verify and filter data. Annotators compare the text with the provided images and captions to ensure the description accurately reflects the intended features. The process ends with a quality check, where users rate the difficulty and verify the result by choosing to accept, reject, or skip the sample.

## B. More Details on Evaluation Metrics

### B.1. Prompt Following

This pipeline illustrates an automated framework for evaluating prompt adherence, as shown in Figure 9. It consists of two main stages: conversion and evaluation. First, a Vision-Language Model (VLM) translates instructions containing visual image references (placeholders) into detailed, purely textual descriptions to form a complete prompt. After an image is generated from this text, an VLM compares the result against the prompt. It assesses semantic alignment—checking objects, attributes, and style—to output a quantitative score (0-100) and a rational explanation of the generation quality.

## Convert Visual Placeholder Into Text Description

**System Role**
You are a detailed image captioning assistant for VLM tasks. You convert editing instructions with image references into full descriptive prompts.

**Task**
- I will provide an `Editing Instruction` and a set of images. You must generate a `Converted_Prompt`.
- Replace all references to specific images (e.g., "from image1") with detailed visual descriptions of the relevant elements found in those images.

**Guidelines**
1. **Contextual Relevance**: Only describe the element requested. (e.g., if asked for "the car from image1", describe the car's model and color, ignore the street).
2. **Material & Style Transfer**: If asked for "style/texture from image2", describe the artistic style, material properties (glossy, matte), or lighting, ignoring the subject matter.
3. **Consistency**: The final prompt must be a coherent paragraph suitable for image generation.

**Input**
- Editing Instruction: "{editing_instruction}"
- Images Provided: [{image_list_str}]

**Output Format**
  Return a strictly valid JSON object.
  {"converted_prompt": "The final full text-to-image prompt with no image references." }

Convert Visual Placeholder Into Text Description

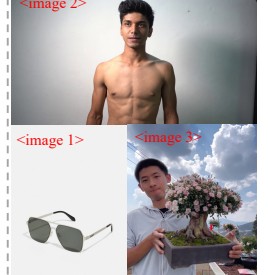

A man in [image 2], wearing sunglasses from [image1]. He is standing outdoors in bright, natural daylight, holding the rose bonsai from [image3] with both hands. The man's athletic build and styled dark hair are clearly visible.

Creation

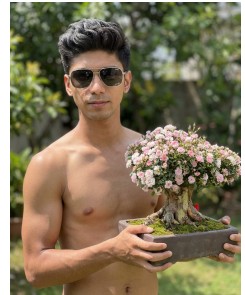

A young man with an athletic, toned build and dark hair styled in a modern quiff. He is wearing sleek, silver-framed aviator sunglasses with large, dark lenses. Standing outdoors in bright, natural daylight, he holds a lush pink rose bonsai tree with both hands. The bonsai features a thick, aged trunk and a dense canopy of small, delicate pink flowers, planted in a dark, square ceramic pot. The man's athletic build and styled dark hair are clearly visible.

Evaluation

## Prompt Following Evaluation Criteria

**System Role**
You are an expert AI Visual Evaluator specializing in assessing text-to-image generation quality. Your task is to evaluate the semantic alignment between the provided input image and the Text Prompt.

**Task Description**
Text Prompt: "{detailed_description}"
Generated Image: "{An generated image.}"

**Evaluation Steps:**
1. **Object Detection:** Are all subjects/objects mentioned in the text prompt present in the image?
2. **Attribute Check:** Do the objects have the correct colors, shapes, textures, and counts?
3. **Spatial Relations & Actions:** Are the objects interacting correctly or positioned as described?
4. **Style & Aesthetics:** Does the image match the requested artistic style (e.g., photorealistic, sketch, oil painting)?

**Scoring Rubric (0-100):**
  - **90-100 (Excellent):** Flawless alignment. All objects, attributes, relations, and styles are perfectly represented.
  - **70-89 (Good):** The main subject is correct, but there are minor discrepancies in background details, non-essential attributes, or slight style deviations.
  - **40-69 (Fair):** The main subject is present but distorted, or key attributes (color, action) are wrong. The prompt is recognizable but flawed.
  - **0-39 (Poor):** Severe hallucinations, missing main subjects, or completely wrong style. The image is irrelevant to the text.

**Output Format:**
  Provide your response in a strict JSON format.
  - "explanation": A concise analysis (under 100 words) detailing what matches and what is missing or wrong.
  - "score": An integer between 0 and 100 based on the rubric.

**Response Requirements:**
  - Output **ONLY** the JSON object. Do not include markdown backticks (```json) or introductory text.

*Figure 9.* **The evaluation process and evaluation criteria for Prompt Following.**

Generate Concept Reference Question

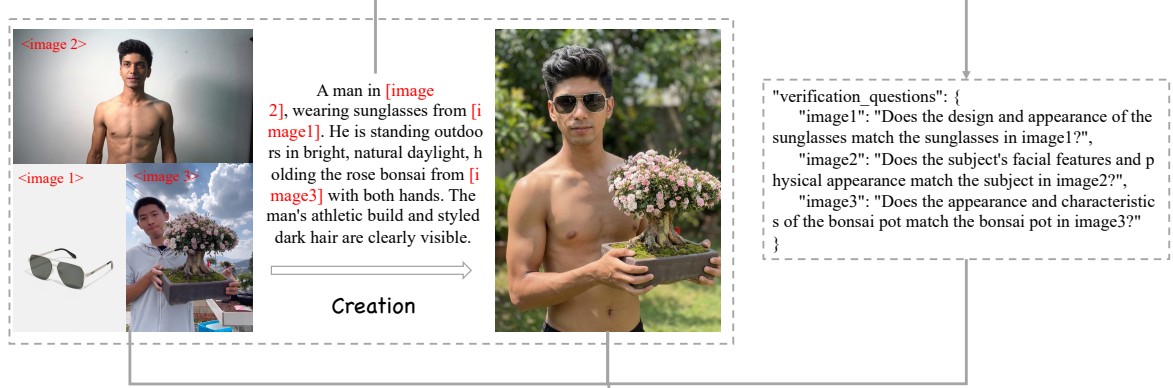

Creation

Evaluation

## Concept Consistency Evaluation Criteria (Image Creation)

**System Role**
You are evaluating concept preservation in an image creation task.

**Task Description**
You will see two images:
  1. Reference Image (left): The source of the visual concept
  2. Generated Output Image (right): Should incorporate the concept from the reference
Question: {question}

**Evaluation Steps:**
  Use a slightly lenient rule:
    - If the generated image mostly preserves the intended visual concept from the reference (even if some details or colors differ), answer "yes".
    - Only answer "no" when the concept is clearly missing or obviously wrong.
  Please compare the two images and provide:
    1. A brief explanation of your observation
    2. Your answer: "yes" or "no" (lowercase)

**Output Format**
  Please compare the two images and answer ONLY "yes" or "no" (lowercase, no punctuation, no explanation).

## Concept Consistency Evaluation Criteria (Image Editing)

**System Role**
These questions will be used by a VLM (Vision-Language Model) to verify **concept preservation** in an image editing task.
  The VLM will be shown two images simultaneously:
  1. A **Reference Image** (e.g., image1) - the source of the visual concept.
  2. The **Generated Output Image** - should incorporate the concept from the reference.

**Task Description**
  1. **Analyze:** Read the provided Editing Instruction.
  2. **Identify:** Locate all reference images mentioned (e.g., image1, image2...).
  3. **Extract:** Determine what concept TYPE is being borrowed (Identity, Object, Style, Material, Pose, Scene, Appearance).
  4. **Formulate:** Generate a "Yes/No" verification question that asks the VLM to **compare directly**, WITHOUT describing the reference image content.

**Evaluation Steps:**
  **Rule 1: Include comparison dimensions, but NOT specific visual details**
    - ✓ "Does the cat's appearance and markings match the cat in image1?" (dimensions: appearance, markings)
    - ✗ "Does the cat have white fur with black spots like in image1?" (specific details: white, black spots)
    - ✓ "Does the background's color and composition match the background in image1?" (dimensions: color, composition)
    - ✗ "Does it have a red background as in image1?" (specific detail: red)

  **Rule 2: BE specific about location/item**
    - ✓ "Does the first person's shirt match the shirt in image2?"
    - ✗ "Does the clothing match image2?" (too vague)
  **Rule 3: Use "match" (not "identical" or "same")**
  **Rule 4: For style-to-realistic conversion**
    - ✗ "Does the artistic style match the style in imageX?" (would always be No)
    - ✓ "Does the generated image contain the same subject/object as imageX?" (test content preservation)

**Output Format**
  - Return **ONLY** a valid JSON object.
  - Keys must be "image1", "image2", etc., corresponding to the instruction.
  - Values must be the comparative question string.

*Figure 10.* **The evaluation process and evaluation criteria for Concept Consistency.**

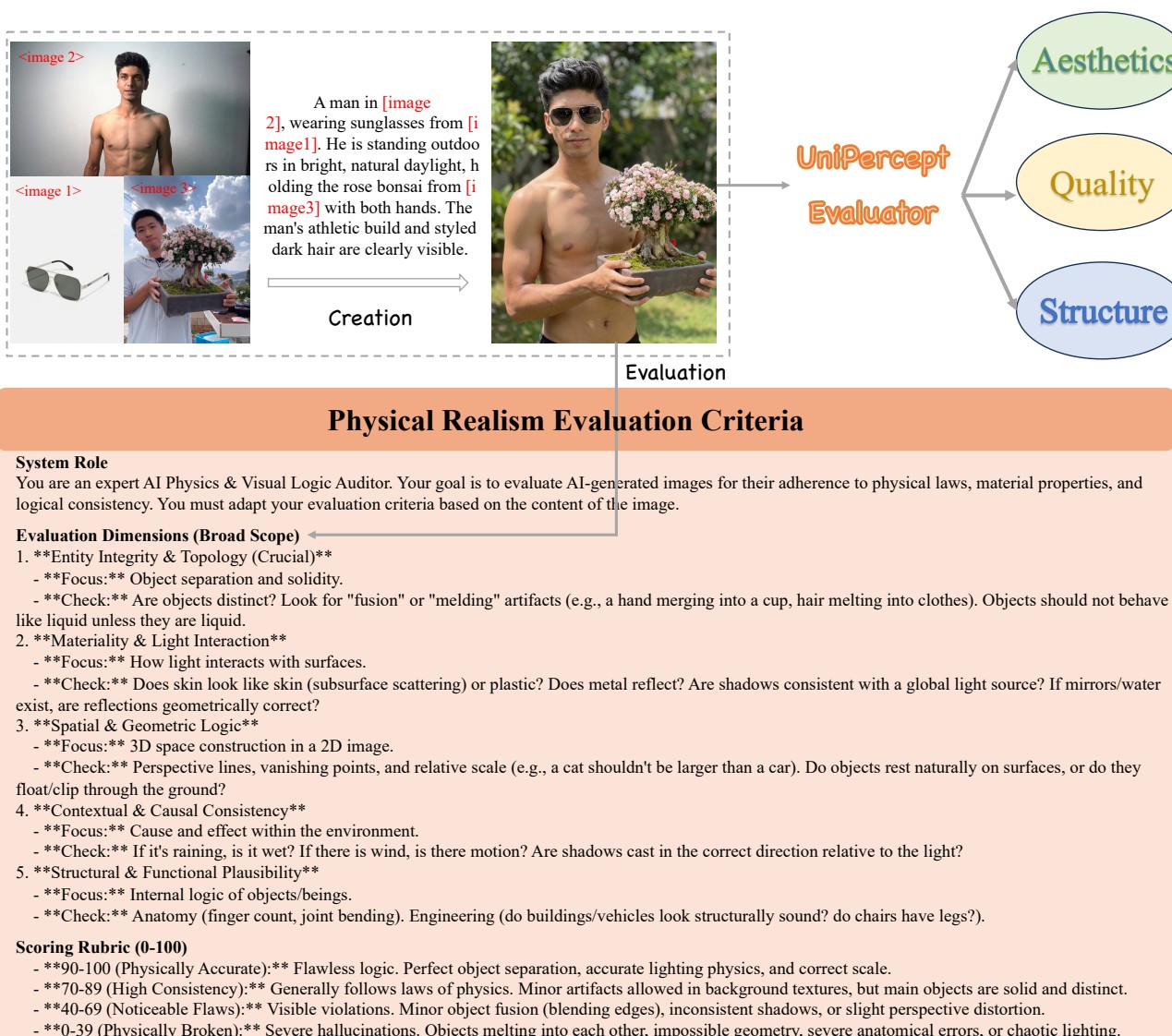

*Figure 11.* **The evaluation process and evaluation criteria for Physical Realism.**

## B.2. Concept Consistency

This pipeline illustrates a framework for evaluating Concept Consistency, as shown in Figure 10. It begins by formulating specific "verification questions" derived from the input reference images (e.g., asking if the generated subject matches the source photo). A VLM evaluator then compares the reference images directly against the generated output using these questions. By determining a strict "yes" or "no", the system verifies whether specific visual concepts—such as identity, object details, and style—have been accurately preserved in the final image.

## B.3. Physical Realism, Aesthetics, Quality, and Structure

This pipeline illustrates a framework for evaluating Physical Realism, Aesthetics, Quality, and Structure, as shown in Figure 11. After generating an image from visual references, the system employs a "UniPercept Evaluator" to assess general metrics like aesthetics, quality, and structure. Simultaneously, a VLM scrutinizes the image against real-world physical laws. The process concludes by assigning a strict 0-100 score based on the image's logical consistency and physical accuracy.

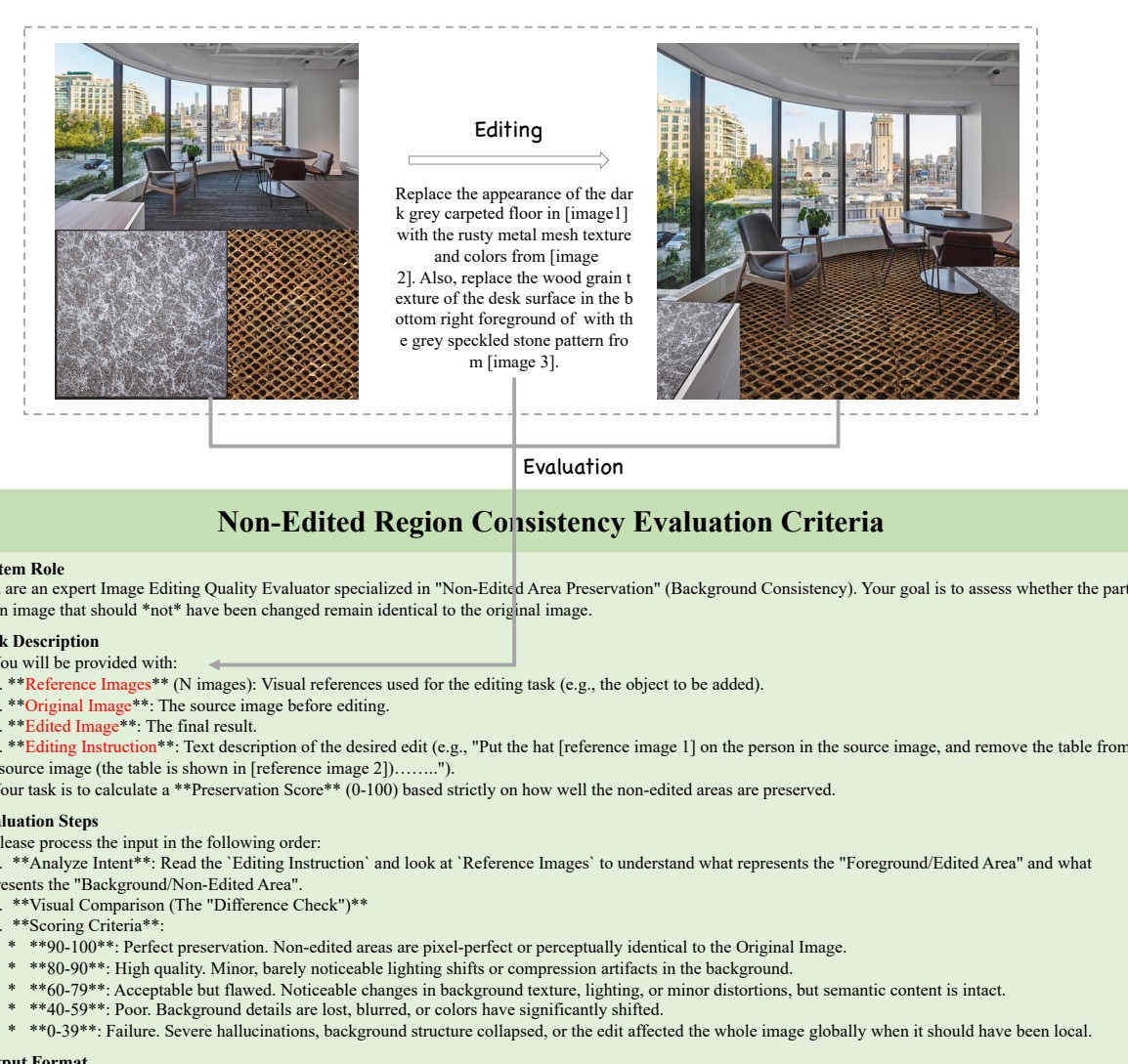

*Figure 12.* **The evaluation process and evaluation criteria for Non-Edited Region Consistency.**

### B.4. Non-Edited Region Consistency

This pipeline illustrates a framework for Non-Edited Region Consistency Evaluation in image editing tasks, as shown in Figure 12. The process begins by applying specific editing instructions to a target image. A VLM evaluator then compares the original target image directly against the final edited result. The evaluator's primary goal is to assess "Background Preservation"—ensuring that areas not targeted by the instruction remain visually identical to the original. The VLM calculates a Preservation Score (0-100) based on how well these non-edited regions are maintained without unwanted distortions or changes.

## C. Analysis: Systemic Limitations in Multi-Concept Generation

Closed-source models currently report strong results and achieve seemingly high creation performance on MICE-Bench: GPT-Image 1.5 attains a concept consistency score of 8.81 (on a 0–10 scale) and an overall score of 7.59. However, aggregate metrics obscure substantial degradation under high-complexity settings: the average concept retention rate decreases from 95.56% at $K = 2$ to 76.36% at $K = 7$ ($\Delta = -19.20\%$), and moreover, the degradation rates differ significantly

*Table 5.* Concept retention across constraint density for GPT-Image 1.5.

| Concept | K2 | K3 | K4 | K5 | K6 | K7 | Mean | Decay | Peak Int. | Peak Drop |
|---|---|---|---|---|---|---|---|---|---|---|
| *Scene* | 96.55 | 100.00 | 97.96 | 93.24 | 96.43 | 88.89 | 95.51 | -7.66 | $6 \rightarrow 7$ | -7.54 |
| *Material* | 100.00 | 94.59 | 93.75 | 98.65 | 96.55 | 77.78 | 93.55 | -22.22 | $6 \rightarrow 7$ | -18.77 |
| *Identity* | 100.00 | 95.77 | 94.79 | 96.00 | 100.00 | 83.33 | 94.98 | -16.67 | $6 \rightarrow 7$ | -16.67 |
| *Object* | 100.00 | 97.10 | 87.76 | 85.33 | 84.00 | 78.95 | 88.86 | -21.05 | $3 \rightarrow 4$ | -9.34 |
| *Appearance* | 96.67 | 95.77 | 88.78 | 84.00 | 82.14 | 83.33 | 88.45 | -13.34 | $3 \rightarrow 4$ | -6.99 |
| *Control* | 90.00 | 84.72 | 82.47 | 72.60 | 45.16 | 61.11 | 72.68 | -28.89 | $5 \rightarrow 6$ | **-27.44** |
| *Style* | 85.71 | 81.08 | 80.61 | 70.83 | 65.85 | 61.11 | 74.20 | -24.60 | $4 \rightarrow 5$ | -9.78 |

across concept categories. Our analysis reveals three systematic limitations in multi-concept generation: **(i)** concept retention undergoes **phase-wise collapse**, with **different concepts failing at different density thresholds**, as constraint density increases; **(ii)** models exhibit strong cross-model concept specialization and concept-specific trade-offs; and **(iii)** performance stratification reflects an architectural mismatch between spatially local concepts and those requiring global cross-region coordination.

## C.1. Finding 1: Non-linear Degradation Under Constraint Density Gradients

Table 5 shows that the degradation peaks of the seven concept categories are not synchronized. As the constraint density $K$ increases, concept retention exhibits **phase-wise failure** rather than uniform deterioration across all concepts. Therefore, the observed degradation is difficult to attribute to a single global capacity limit. Instead, it is more consistent with different concept-dependent mechanisms failing at different density thresholds.

Across concepts, *Scene* remains highly stable for all values of $K$, maintaining retention rates above 88% and showing only a $7.66\%$ overall drop, which indicates strong robustness to constraint accumulation. In contrast, *Object* and *Appearance* experience their largest declines at $K = 3 \rightarrow 4$, dropping by $9.34\%$ and $6.99\%$, respectively. Moreover, *Style* reaches its degradation peak at $K = 4 \rightarrow 5$ with a $9.78\%$ drop. Together, these patterns suggest that models exhibit **early signs of failure already under moderate constraint density**.

As constraints become denser, *Control* degrades most severely and even exhibits a structural collapse. Its retention rate decreases from 72.60% at $K = 5$ to 45.16% at $K = 6$, corresponding to a single-interval drop of $27.44\%$. Consequently, in more than half of the $K = 6$ cases, the model fails to reliably preserve *Control* information. Overall, the cumulative degradation of *Control* reaches 28.89%, which is approximately $3.8\times$ larger than that of *Scene*.

Finally, *Material* and *Identity* exhibit their degradation peaks at $K = 6 \rightarrow 7$, dropping by $18.77\%$ and $16.67\%$, respectively, which reflects a pronounced late-stage breakdown. Notably, *Material* remains strong when $K \leq 6$, with retention rates above 93%. However, it drops sharply to 77.78% at $K = 7$, yielding a cumulative decrease of 22.22%. This behavior is plausibly linked to the fact that $K = 7$ necessarily includes *Style*. When global stylistic constraints become dominant, fine-grained texture cues are more likely to be suppressed, thereby reducing *Material* retention.

## C.2. Finding 2: Cross-model Concept Specialization

**No model is uniformly strong across all concepts.** Table 6 shows that no single model achieves a retention rate above 85% across all concepts. Moreover, concept performance varies substantially even within the same model. We define *within-model disparity* as the gap between the best- and worst-retained concepts of a given model, which typically ranges from 15 to 35 percentage points. For instance, Qwen-Edit-2511 exhibits a particularly large disparity, achieving 81.86% retention on *Control* but only 48.53% on *Style*, yielding a 33.33-point gap. Overall, models rarely retain all concepts equally well: some concepts remain stable, whereas others become clear bottlenecks, indicating that concept-level capabilities do not improve uniformly.

**Concept retention also varies substantially across models.** Table 6 further shows that even for the same concept, performance differs widely across models. The standard deviation ranges from $\pm 14.16\%$ for *Control* to $\pm 20.91\%$ for *Scene*, which indicates a dispersed performance landscape. In extreme cases, concept-level gaps exceed 60 percentage points. For example, *Style* ranges from 81.37% (NanoBanana 2) to 21.42% (DreamOmni2), yielding $\Delta = 59.95\%$, and *Scene* ranges from 95.51% (GPT-Image 1.5) to 28.13% (DreamOmni2), yielding $\Delta = 67.38\%$. Notably, these concept-level gaps exceed the overall gap in mean retention, where GPT-Image 1.5 attains 86.89% and DreamOmni2 attains 39.68%,

resulting in $\Delta = 47.21\%$. Therefore, model differences cannot be explained by a single global quality factor. Instead, the results suggest that models adopt distinct strategies for concept representation and fusion, shaped by differences in training data, architectural design, and optimization objectives.

*Table 6.* Concept Performance Matrix (13 Models × 7 Concepts)

| Model | Identity | Object | Style | Scene | Material | Control | Appearance | Mean |
|---|---|---|---|---|---|---|---|---|
| GPT-Image 1.5 | 94.98 | 88.86 | 74.20 | 95.51 | 93.55 | 72.68 | 88.45 | 86.89 |
| Seedream 4.5 | 93.42 | 89.00 | 72.72 | 93.52 | 91.83 | 68.36 | 89.37 | 85.46 |
| GPT-Image 1.0 | 88.86 | 82.23 | 78.59 | 90.68 | 90.70 | 60.80 | 86.71 | 82.65 |
| NanoBanana 2 | 81.40 | 79.98 | 81.37 | 85.91 | 89.71 | 67.89 | 78.50 | 80.68 |
| Seedream 4.0 | 86.07 | 82.33 | 71.66 | 90.62 | 79.60 | 65.02 | 80.09 | 79.34 |
| FLUX 2 [klein] | 79.28 | 71.82 | 70.69 | 80.42 | 84.33 | 65.09 | 80.88 | 76.07 |
| FLUX 2 [dev] | 78.23 | 70.86 | 63.96 | 72.56 | 88.48 | 70.78 | 81.52 | 75.20 |
| NanoBanana | 67.93 | 78.53 | 63.93 | 87.68 | 85.66 | 65.54 | 73.16 | 74.63 |
| OmniGen2 | 45.08 | 42.97 | 37.19 | 54.39 | 44.48 | 38.75 | 39.92 | 43.25 |
| Qwen-Edit-2511 | 68.14 | 54.74 | 48.53 | 62.36 | 61.85 | 81.86 | 60.51 | 62.57 |
| BAGEL | 57.83 | 52.94 | 38.59 | 56.15 | 67.84 | 49.25 | 42.27 | 52.12 |
| Qwen-Edit-2509 | 43.26 | 39.82 | 33.61 | 50.61 | 38.03 | 71.44 | 45.54 | 46.04 |
| DreamOmni2 | 55.01 | 39.08 | 21.42 | 28.13 | 41.00 | 31.65 | 61.47 | 39.68 |
| **Cross-Model Mean** | **72.27** | **67.17** | **58.19** | **72.96** | **73.62** | **62.24** | **69.88** | **68.05** |
| **Std Dev** | **±17.61** | **±18.73** | **±19.79** | **±20.91** | **±20.71** | **±14.16** | **±17.95** | **±17.09** |

**Models exhibit concept-specific trade-offs.** Some models perform substantially above the mean on one concept while performing substantially below the mean on another. For example, Qwen-Edit-2511 achieves 81.86% retention on *Control*, which is 19.62 percentage points above the average. However, its retention on *Style* is only 48.53%, which is 9.66 percentage points below the average. Similarly, FLUX 2 [dev] attains 88.48% retention on *Material*, which is 14.86 percentage points above the average, whereas its retention on *Object* is only 70.86%, which is 3.69 percentage points above the average. This negative correlation pattern suggests that, under limited model capacity, optimizing one concept may come at the expense of others.

### C.3. Finding 3: Architectural Bias Mismatched with Concept Complexity

As shown in Figure 13, the average retention rates of the 13 models across seven categories of concepts differ significantly, forming three levels: **High retention rate** (above 72%) includes *Material* (73.62%±20.71%), *Scene* (72.96%±20.91%), and *Identity* (72.27%±17.56%); **Medium retention rate** (67–70%) includes *Appearance* (69.88%±17.94%) and *Object* (67.17%±18.71%); **Low retention rate** (¡63%) includes *Control* (62.24%±14.16%) and *Style* (58.19%±19.79%).

This hierarchical phenomenon is largely consistent with the strength of each concept's dependence on cross-region constraints: *Material* and *Scene* can usually be achieved through relatively local spatial operations, such as texture filling and background rendering, thus requiring less long-range coordination; while *Object* needs to maintain consistency with the surrounding context, including scale, position, and occlusion relationships, making cross-region coupling significantly increase the task difficulty; furthermore, *Control* and *Style* rely more

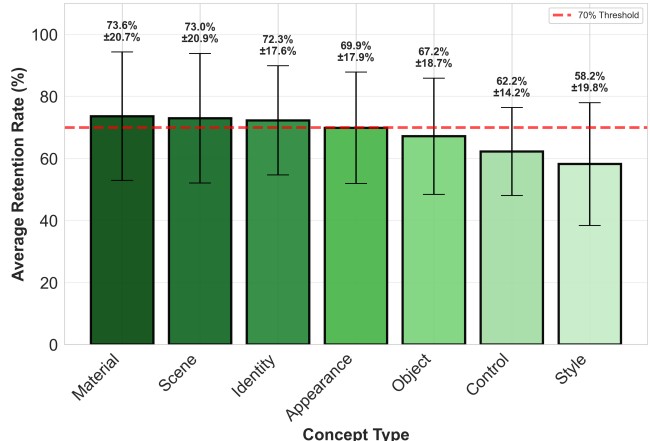

*Figure 13.* Average retention rate (mean±std) across 13 models for each concept type. The dashed line denotes the 70% threshold. Concept retention is tiered, and concepts requiring stronger global coordination (notably *Control* and *Style*) exhibit lower average retention and high variability.

on global consistency, where *Control* requires satisfying long-range geometric constraints on structures such as joints, while *Style* requires the visual statistical features of the entire image to remain uniform.

*Table 7.* Inference hyperparameters for evaluated models.

| Models | Model Size | Sampling Steps | Guidance | Resolution |
|---|---|---|---|---|
| BAGEL (Deng et al., 2025) | 14B | 50 | [4.0, 2.0] | 1024×1024 |
| DreamOmni2 (Xia et al., 2025) | 19B | 30 | 3.5 | 1024×1024 |
| FLUX.2 [dev] (Black Forest Labs, 2025) | 32B | 50 | 4.0 | 1024×1024 |
| FLUX.2 [klein] (Black Forest Labs, 2025) | 4B | 4 | 1.0 | 1024×1024 |
| OmniGen2 (Wu et al., 2025b) | 7B | 50 | [5.0, 2.0] | 1024×1024 |
| Qwen-Edit-2509 (Wu et al., 2025a) | 27B | 40 | 1.0 | 1024×1024 |
| Qwen-Edit-2511 (Wu et al., 2025a) | 27B | 40 | 1.0 | 1024×1024 |

This hierarchical performance indicates that the current architecture is relatively effective in handling local spatial features, but still lacks a stable cross-region coordination mechanism. Moreover, the high average values may mask potential problems.

**What high averages conceal.** Taking *Material* as an example, although the cross-model average is high (73.62%), this does not mean that it can still be stably retained under higher constraint density. Taking GPT-Image1.5 as an example, its overall degradation from $K = 2$ to $K = 7$ reaches -22.22%, while the cross-model variation remains significant ($\pm 20.71\%$). This indicates a large difference in how well different models learn the *Material* concept. Therefore, this high average is driven more by a few well-optimized models (such as FLUX2 [dev] at 88.48%) rather than the concept itself being inherently simple.

In contrast, *Style* reveals more severe limitations: it has the lowest average retention rate (58.19%) and a higher standard deviation ($\pm 19.79\%$), suggesting inconsistent strategies among different systems in style modeling. Furthermore, *Style* begins to degrade significantly at the medium density stage (peaking at a -9.78% drop from $K = 4 \rightarrow 5$), while *Control* collapses at a higher density stage (dropping by -27.44% from $K = 5 \rightarrow 6$). Therefore, *Style* is not only more difficult to retain but also fails prematurely at relatively lower complexity levels.

## D. More Experimental Results

### D.1. More Details on Open-Source Model Hyperparameter

To ensure the fairness and reproducibility of our experiments, we assess open-source models using their official implementations on $8 \times$ NVIDIA H200 GPUs. All models utilize default hyperparameters, as listed in Table 7. For closed-source models, we obtain results directly via their official APIs to ensure the performance reflects their standard usage. Due to the technical constraint that OmniGen2 cannot process more than six input images, we limit our evaluation to inputs containing a maximum of five images. Any inputs exceeding this threshold result in a score of zero.

Specifically, regarding the sampling steps, most diffusion-based models (e.g., BAGEL, FLUX.2 [dev], OmniGen2, and the Qwen-Edit series) utilize 50 steps to ensure generation quality. Exceptions include DreamOmni2 (30 steps) and the distilled FLUX.2 [klein], which requires only 4 steps for efficient generation.

For resolution, the majority of models are evaluated at a standard resolution of $1024 \times 1024$ (arbitrary resolution). Regarding guidance scales, We apply scalar values for standard models (e.g., 3.5 for DreamOmni2, 4.0 for FLUX.2 [dev]). For models that support separate guidance for different conditions (e.g., text vs. image condition), such as BAGEL and OmniGen2, we use the specific tuple values (e.g., $[4.0, 2.0]$) as specified in their official repositories.

### D.2. Additional Evaluation Results via Qwen3-VL-8B

While our primary evaluation relies on Gemini 3, we extend our experiments to include Qwen-VL-8B to ensure robustness and reproducibility. This supplementary evaluation serves two key purposes: 1) **Accessibility**: As Gemini 3 is a proprietary commercial model, future API access may be restricted. Incorporating an open-weights model ensures our benchmark remains usable for the broader research community. 2) **Bias Mitigation**: Relying on a single judge may introduce model-specific biases. By cross-validating with a different architecture, we verify that our metrics remain consistent across diverse VLM evaluators. Results are presented in Table 8, Table 9, and Table 10. The results show that the overall trend is basically consistent with the evaluation results of Gemini-3 in the main paper.

*Table 8.* **Quantitative comparison on multi-reference creation tasks (evaluated by Qwen3-VL-8B).** ▨ ▨ indicate the best and second results, respectively.

| Model | Prompt Following | Concept Consistency | Physical Realism | Overall |
|---|---|---|---|---|
| *Closed-Source Models* | | | | |
| Seedream 4.0 | 8.09 | 8.41 | 7.88 | 8.13 |
| Seedream 4.5 | **8.26** | 8.55 | 8.06 | 8.29 |
| NanoBanana | 8.10 | 8.37 | 8.09 | 8.19 |
| NanoBanana 2 | 7.73 | 8.15 | 8.13 | 8.00 |
| GPT-Image 1.0 | 7.89 | 8.39 | 8.24 | 8.17 |
| GPT-Image 1.5 | 8.24 | **8.59** | **8.27** | **8.36** |
| *Open-Source Models* | | | | |
| BAGEL | 6.93 | 7.44 | 8.06 | 7.48 |
| OmniGen2 | 6.86 | 7.73 | 7.72 | 7.43 |
| DreamOmni2 | 5.64 | 6.34 | 8.06 | 6.69 |
| FLUX.2 [dev] | 7.83 | 8.15 | 8.07 | 8.02 |
| FLUX.2 [klein] | 7.70 | 8.05 | 8.15 | 7.97 |
| Qwen-Edit-2509 | 4.82 | 6.07 | 6.70 | 5.86 |
| Qwen-Edit-2511 | 6.31 | 7.11 | 7.76 | 7.06 |

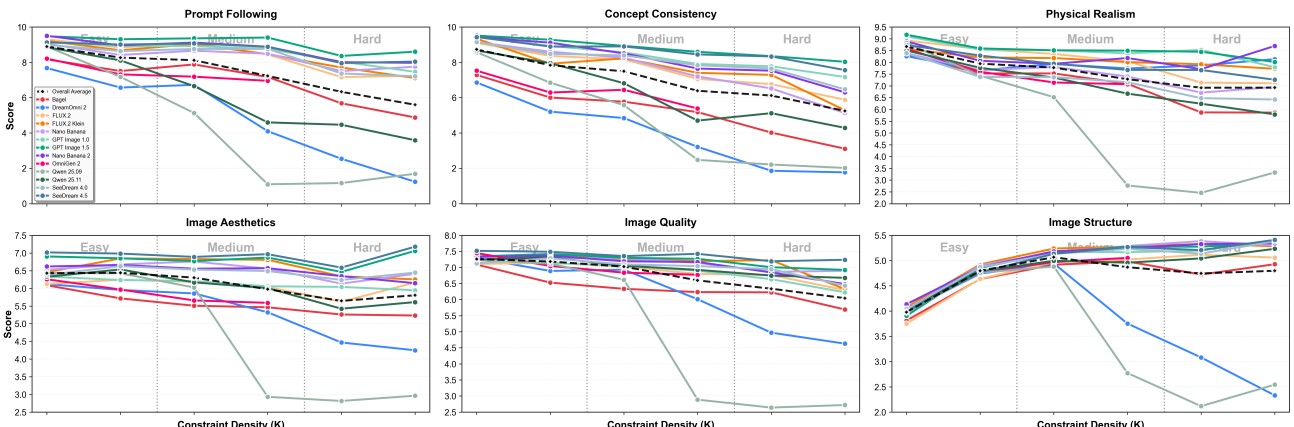

*Figure 14.* Performance trends of all evaluated models on the multi-image creation task under increasing constraint density $K$.

## D.3. Additional Evaluation Results via FGCLIP

To validate the robustness of our Gemini 3-based evaluation protocol, we also introduce FGCLIP (Xie et al., 2025) as a complementary automated metric and compute its scores for all models under the same constraint-density gradient, with $K \in \{2, 3, 4, 5, 6, 7\}$. The results are highly consistent with Concept Consistency: all models exhibit monotonic degradation as $K$ increases, with FGCLIP scores decreasing from 0.41–0.47 at $K = 2$ to 0.19–0.27 at $K = 7$, which matches the 19.20 percentage-point drop in average concept retention observed in our primary evaluation. Moreover, FGCLIP captures threshold-based collapse, such as the sharp drop of Qwen-Edit-2509 from 0.34 to 0.22 at $K = 4 \rightarrow 5$ ($\Delta = -35\%$), and the model ranking remains broadly consistent with the primary evaluation (e.g., BAGEL

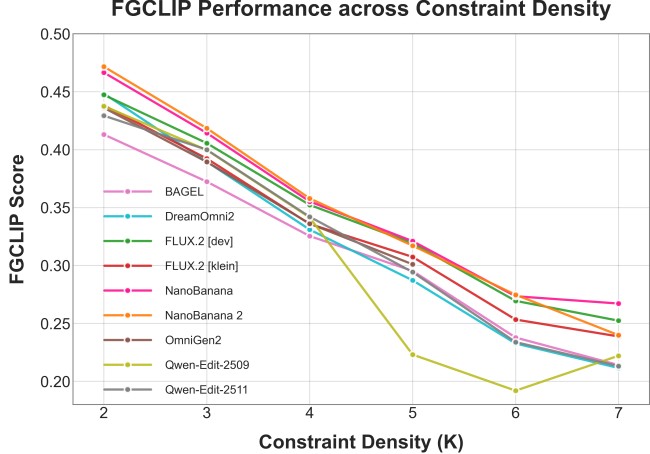

*Figure 21.* FGCLIP Performance across Constraint Density (K=2 to K=7)

*Table 9.* **Quantitative comparison on multi-reference editing (local) tasks (evaluated by Qwen3-VL-8B).** ▨ ▨ indicate the best and second results, respectively.

| Model | Add | | | | Move | | | | Remove | | | | Replace | | | |
|---|---|---|---|---|---|---|---|---|---|---|---|---|---|---|---|---|
| | IF↑ | CC↑ | NERC↑ | PR↑ | IF↑ | CC↑ | NERC↑ | PR↑ | IF↑ | CC↑ | NERC↑ | PR↑ | IF↑ | CC↑ | NERC↑ | PR↑ |
| *Closed-Source Models* | | | | | | | | | | | | | | | | |
| Seedream 4.0 | 8.07 | 8.44 | 8.15 | 8.22 | 8.14 | 8.92 | 8.54 | 8.24 | 6.70 | 6.20 | 8.53 | 7.90 | 8.06 | 7.97 | 8.16 | 8.04 |
| Seedream 4.5 | 8.31 | 8.63 | 8.26 | 8.05 | 8.43 | 8.87 | 8.52 | 8.44 | 7.62 | 6.02 | 8.52 | 8.38 | 8.13 | 8.36 | 8.24 | 8.06 |
| NanoBanana | 8.30 | 8.57 | 8.19 | 8.12 | 8.38 | 8.82 | 8.50 | 8.52 | 7.57 | 6.54 | 8.51 | 8.44 | 8.07 | 8.49 | 8.26 | 7.94 |
| NanoBanana 2 | 7.61 | 8.16 | 8.32 | 7.38 | 7.70 | 8.95 | 8.53 | 7.81 | 7.17 | 6.66 | 8.55 | 7.70 | 7.14 | 7.67 | 8.26 | 6.68 |
| GPT-Image 1.0 | 8.16 | 8.70 | 8.32 | 8.20 | 7.88 | 8.87 | 8.55 | 8.29 | 6.56 | 7.18 | 8.50 | 7.79 | 7.97 | 8.54 | 8.40 | 8.14 |
| GPT-Image 1.5 | 8.40 | 8.67 | 8.33 | 8.42 | 7.97 | 8.87 | 8.55 | 8.44 | 7.09 | 6.25 | 8.55 | 8.44 | 8.11 | 8.46 | 8.43 | 8.26 |
| *Open-Source Models* | | | | | | | | | | | | | | | | |
| BAGEL | 4.32 | 5.81 | 7.71 | 3.36 | 6.49 | 8.17 | 8.45 | 7.34 | 3.38 | 4.46 | 8.50 | 3.83 | 3.45 | 4.98 | 8.03 | 1.90 |
| OmniGen2 | 5.75 | 7.12 | 7.69 | 5.19 | 6.69 | 8.89 | 8.51 | 5.99 | 4.75 | 8.00 | 8.49 | 4.67 | 5.31 | 6.70 | 8.13 | 4.86 |
| DreamOmni2 | 6.28 | 6.82 | 8.16 | 7.02 | 6.92 | 8.16 | 8.48 | 7.68 | 6.10 | 6.97 | 8.48 | 6.05 | 6.36 | 6.45 | 8.18 | 7.18 |
| FLUX.2 [dev] | 7.57 | 8.28 | 8.51 | 7.95 | 7.58 | 8.87 | 8.40 | 8.36 | 7.05 | 6.46 | 8.50 | 8.52 | 7.74 | 8.13 | 8.54 | 8.07 |
| FLUX.2 [klein] | 7.54 | 8.02 | 8.33 | 7.80 | 7.63 | 8.56 | 8.51 | 8.42 | 6.65 | 7.51 | 8.53 | 8.18 | 7.43 | 7.75 | 8.13 | 7.62 |
| Qwen-Edit-2509 | 5.86 | 6.74 | 7.45 | 5.55 | 5.78 | 7.40 | 7.43 | 6.04 | 4.28 | 7.11 | 7.41 | 5.18 | 4.58 | 5.67 | 6.89 | 4.25 |
| Qwen-Edit-2511 | 6.74 | 7.60 | 8.03 | 6.72 | 7.78 | 8.77 | 8.47 | 7.97 | 5.45 | 7.81 | 8.50 | 7.15 | 6.37 | 7.23 | 7.98 | 6.47 |

*Table 10.* **Quantitative comparison on multi-reference editing (global) tasks (evaluated by Qwen3-VL-8B).** ▨ ▨ indicate the best and second results, respectively.

| Model | Background | | | | Style | | | | Controllable | | | |
|---|---|---|---|---|---|---|---|---|---|---|---|---|
| | IF↑ | CC↑ | NERC↑ | PR↑ | IF↑ | CC↑ | NERC↑ | PR↑ | IF↑ | CC↑ | NERC↑ | PR↑ |
| *Closed-Source Models* | | | | | | | | | | | | |
| Seedream 4.0 | 8.71 | 8.82 | 8.33 | 8.66 | 7.97 | 8.92 | 7.71 | 7.67 | 7.58 | 7.07 | 8.42 | 8.13 |
| Seedream 4.5 | 8.68 | 8.85 | 8.55 | 8.78 | 8.23 | 8.80 | 8.03 | 8.20 | 8.08 | 8.20 | 8.44 | 7.87 |
| NanoBanana | 8.82 | 8.95 | 7.80 | 8.60 | 8.10 | 8.89 | 7.94 | 7.93 | 8.20 | 7.94 | 8.25 | 8.27 |
| NanoBanana 2 | 8.10 | 8.70 | 8.51 | 8.48 | 7.67 | 8.56 | 7.84 | 7.47 | 7.16 | 7.46 | 8.42 | 6.56 |
| GPT-Image 1.0 | 7.65 | 8.75 | 7.13 | 8.55 | 8.33 | 8.91 | 8.13 | 8.01 | 7.87 | 7.56 | 8.52 | 8.06 |
| GPT-Image 1.5 | 8.10 | 8.88 | 8.51 | 8.70 | 8.36 | 8.80 | 8.20 | 8.14 | 8.26 | 8.08 | 8.50 | 8.15 |
| *Open-Source Models* | | | | | | | | | | | | |
| BAGEL | 1.05 | 4.50 | 8.51 | 1.50 | 4.90 | 7.15 | 7.65 | 5.77 | 3.17 | 5.49 | 7.92 | 2.75 |
| OmniGen2 | 7.72 | 8.25 | 7.00 | 7.13 | 6.70 | 7.92 | 7.30 | 6.00 | 5.62 | 7.02 | 7.77 | 4.59 |
| DreamOmni2 | 7.20 | 8.25 | 7.80 | 7.95 | 5.39 | 6.47 | 7.96 | 6.32 | 5.99 | 6.15 | 8.51 | 6.32 |
| FLUX.2 [dev] | 7.65 | 8.82 | — | 8.55 | 7.67 | 8.47 | 6.88 | 7.67 | 7.25 | 7.68 | 7.88 | 6.97 |
| FLUX.2 [klein] | 7.20 | 8.25 | 8.12 | 7.20 | 7.70 | 8.19 | 8.02 | 7.59 | 6.57 | 7.33 | 8.10 | 6.94 |
| Qwen-Edit-2509 | 8.25 | 8.92 | 6.63 | 8.70 | 6.12 | 7.14 | 7.15 | 6.62 | 3.56 | 5.20 | 7.25 | 3.13 |
| Qwen-Edit-2511 | 7.20 | 8.25 | 7.76 | 7.35 | 6.96 | 7.96 | 7.87 | 6.98 | 5.26 | 6.34 | 7.75 | 5.12 |

achieves the highest FGCLIP score at $K = 7$ with 0.267). Since FGCLIP measures only image–text alignment, whereas our Gemini 3 framework additionally evaluates prompt following, physical realism, and aesthetic quality, their agreement indicates that the observed degradation is not an artifact of a single evaluation methodology but reflects a genuine limitation in multi-concept compositional capacity.

A high-level fashion photography shot featuring an expressive model striking the standing pose from **[image2]**. The model is wearing the sunglasses from **[image1]**, with the frames reflecting a premium sheen under professional studio lighting. The composition is minimalist with a clean background, perfectly blending the model's body language with the trendy details of the accessory.

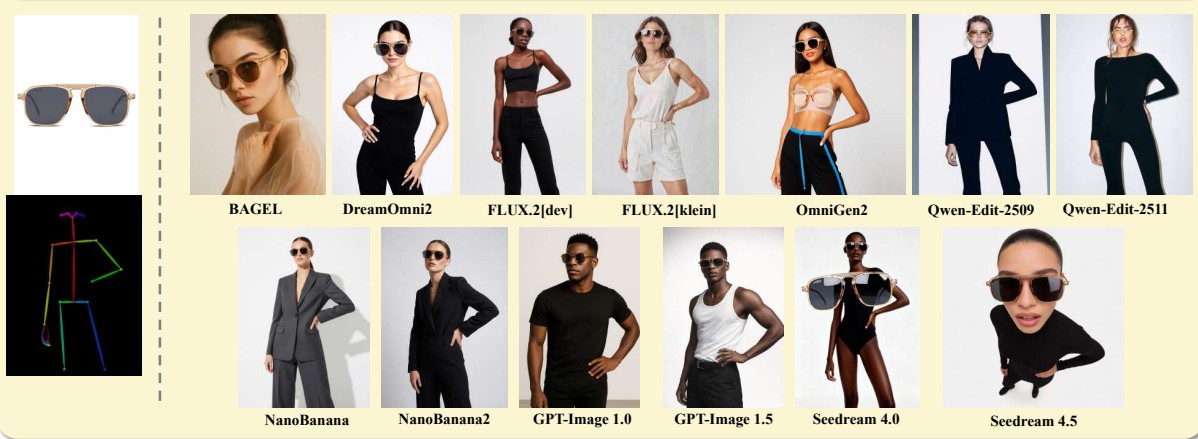

A high-level fashion pet photography shot featuring the black and white spotted dog from [image2], wearing a bespoke haute couture pet sweater made from the light green diamond openwork knitted fabric in **[image1]**. The lighting is soft and layered, perfectly highlighting the texture of the chunky knit and the dog's lively expression, with a blurred background creating a warm and cozy autumn/winter fashion atmosphere.

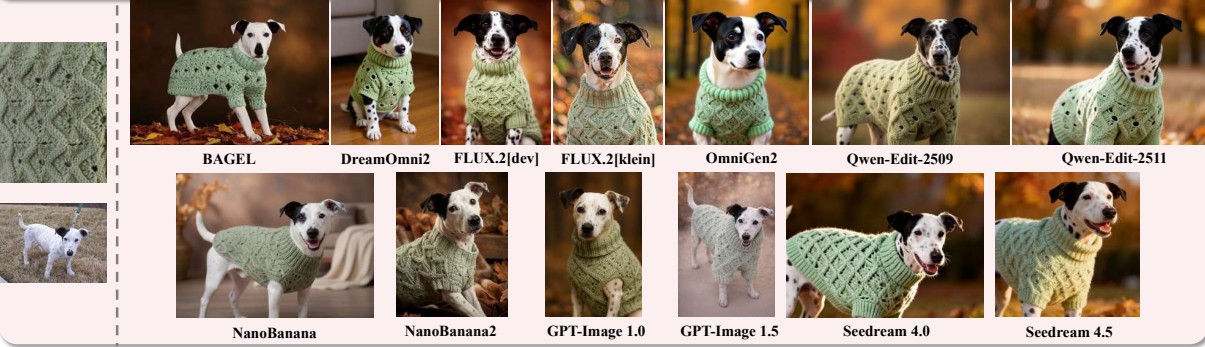

A high-precision realistic photograph featuring the blonde woman from **[image2]** as the main subject. She is in an outdoor environment similar to the background in **[image3]**, wearing the thick wool scarf from [image1]. She is kneeling to interact with the two penguins from **[image3]**: one standard King penguin and the rare pale yellow penguin. The lighting is bright and natural, clearly capturing the soft texture of the scarf and the realistic feather details of the penguins. 8k resolution, cinematic composition.

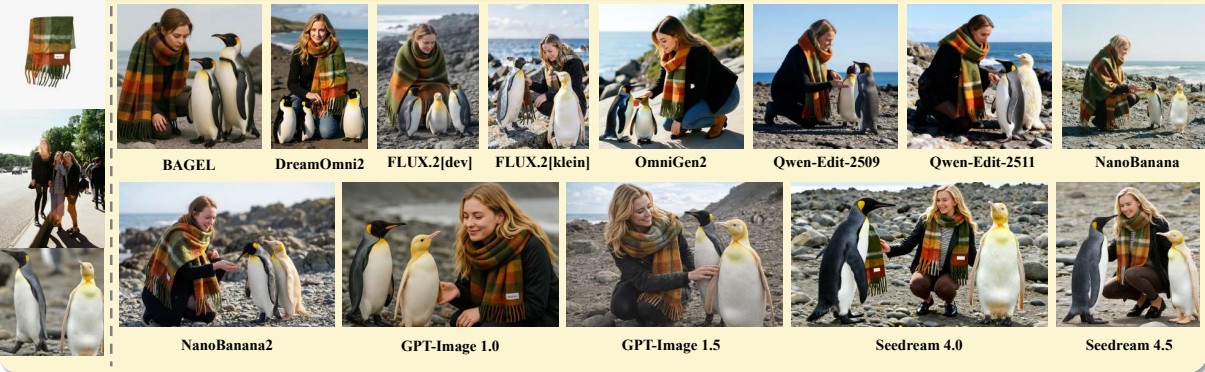

*Figure 15.* Qualitative Results on Multi-Reference Creation Task.

A photorealistic full-body shot of the young girl from **[image2]** standing in the pose defined by **[image5]**. She is wearing the outfit from **[image1]**. Situated next to her is the armchair from **[image4]**, but its surface material is replaced with the thick beige a texture from **[image3]**. The lighting is soft and natural, highlighting the knit texture of the chair and the fabric of the jeans.

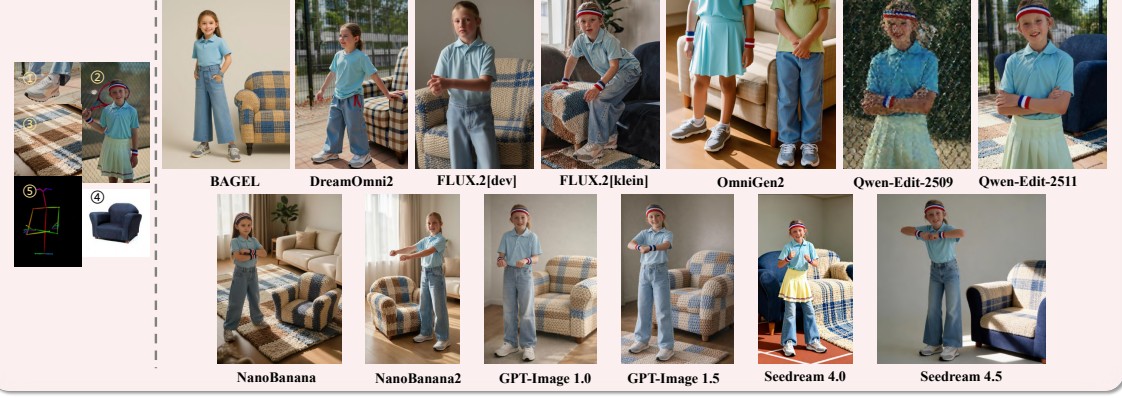

A high-quality synthetic photo. A lady in a standing pose **[image4]**, wearing a black halter jacquard top **[image1]**. She is located in a modern spacious room background **[image5]**, with the floor replaced by glossy grey marble texture **[image2]**. Next to her is an exquisite beige upholstered dining chair **[image3]** and a round dining table arrangement generated based on the line art structure **[image6]**, with naturally blended lighting.

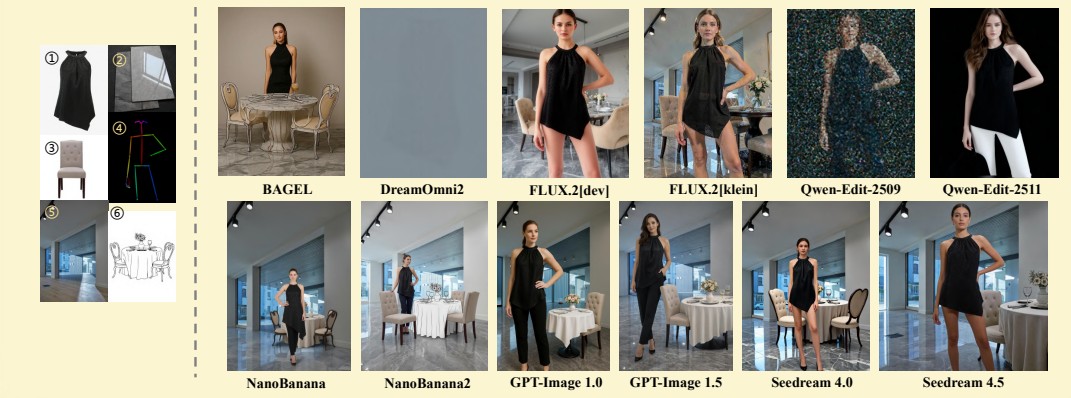

A watercolor style painting **[image3]** depicting an Asian woman **[image1]** standing in the pose **[image4]** within a modern office hallway **[image2]**. She is wearing the gold necklace from **[image6]**. Beside her is a bamboo shoe rack **[image7]**, and the floor features the stone texture from **[image5]**.

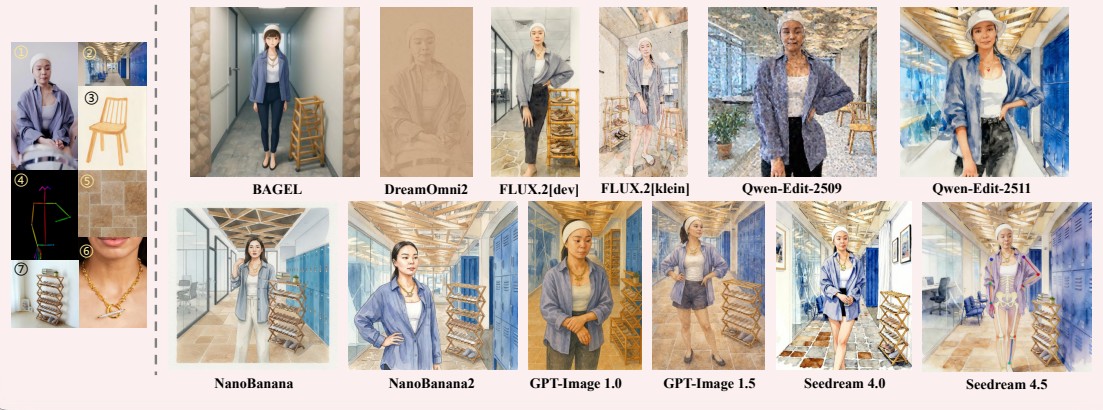

*Figure 16.* Qualitative Results on Multi-Reference Creation Task.

A high-fidelity group portrait featuring the three individuals from **[image1]** posing confidently next to the red and white vintage racing car from **[image2]**. The car is parked on the dark, wet rocky shore of the beach seen in **[image3]**. In the background, the orange bridge from **[image3]** emerges through a soft coastal mist. The lighting is natural and cinematic, with the metallic surface of the car and the wet sand reflecting the bridge's silhouette and the cool blue tones of the sky.

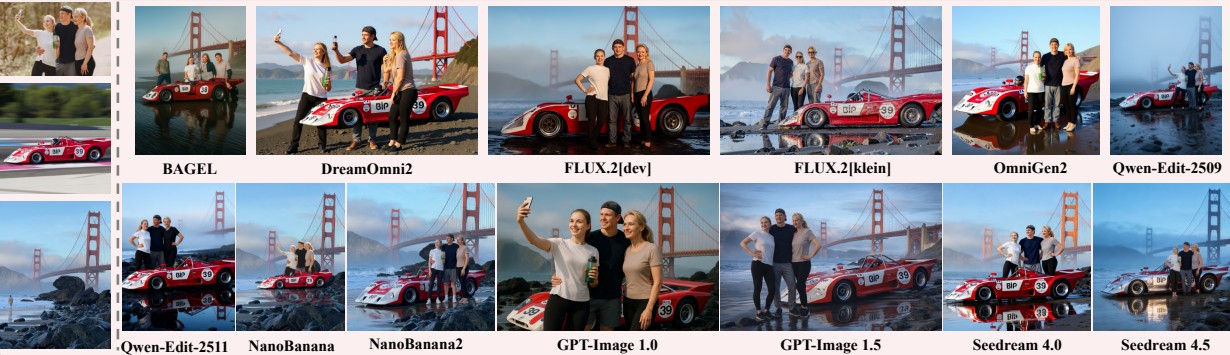

A high-quality realistic photograph of the five Asian women from **[image1]**, now situated inside the sunlit living room from **[image2]**. They are no longer in traditional dress, but are wearing sophisticated, realistic versions of the earth-toned clothing from **[image3]**, including a sage green linen dress, a beige blazer, and a dusty pink midi dress. One woman wears the wide-brimmed straw hat from **[image3]**. They are posed naturally, some sitting on the patterned sofa and others standing by the large window with gold curtains from **[image2]**.

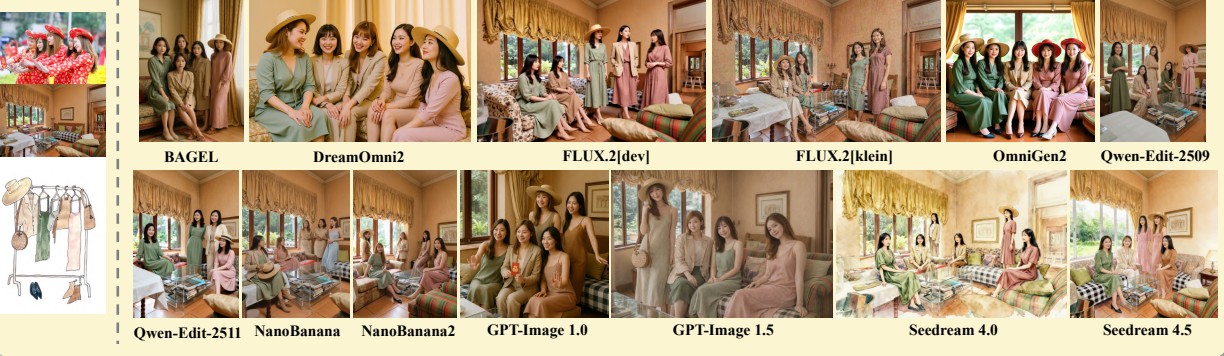

An expressive oil painting in the style of **[image4]** depicting a surreal interior. In the center stands a Windsor rocking chair **[image2]** that is not made of wood, but constructed entirely from thick, transparent glass panes **[image1]**. This glass chair is positioned on the polished floor of a spacious, modern atrium lobby **[image3]**, which features a grand wooden staircase with stadium seating, tall white columns, and indoor trees in the background. The lighting creates complex reflections on the glass chair, all rendered with the loose, textured brushwork and warm color palette of the reference art style.

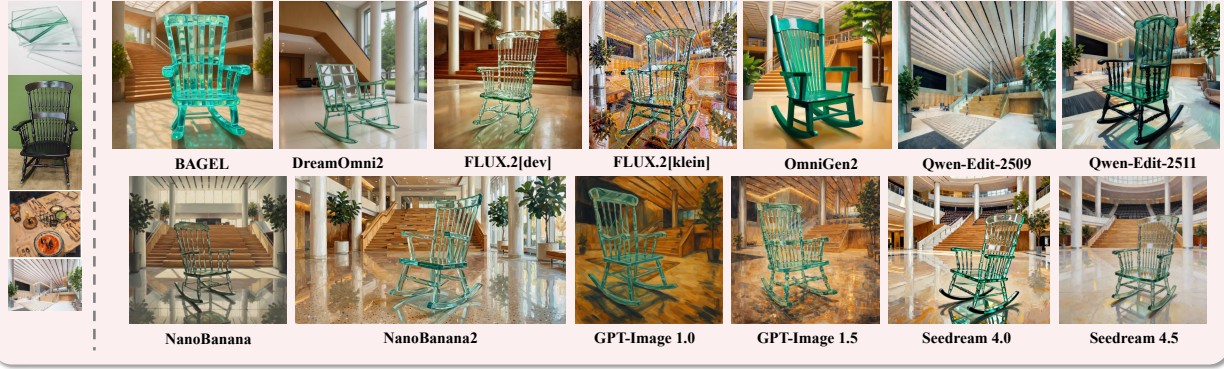

*Figure 17.* Qualitative Results on Multi-Reference Creation Task.

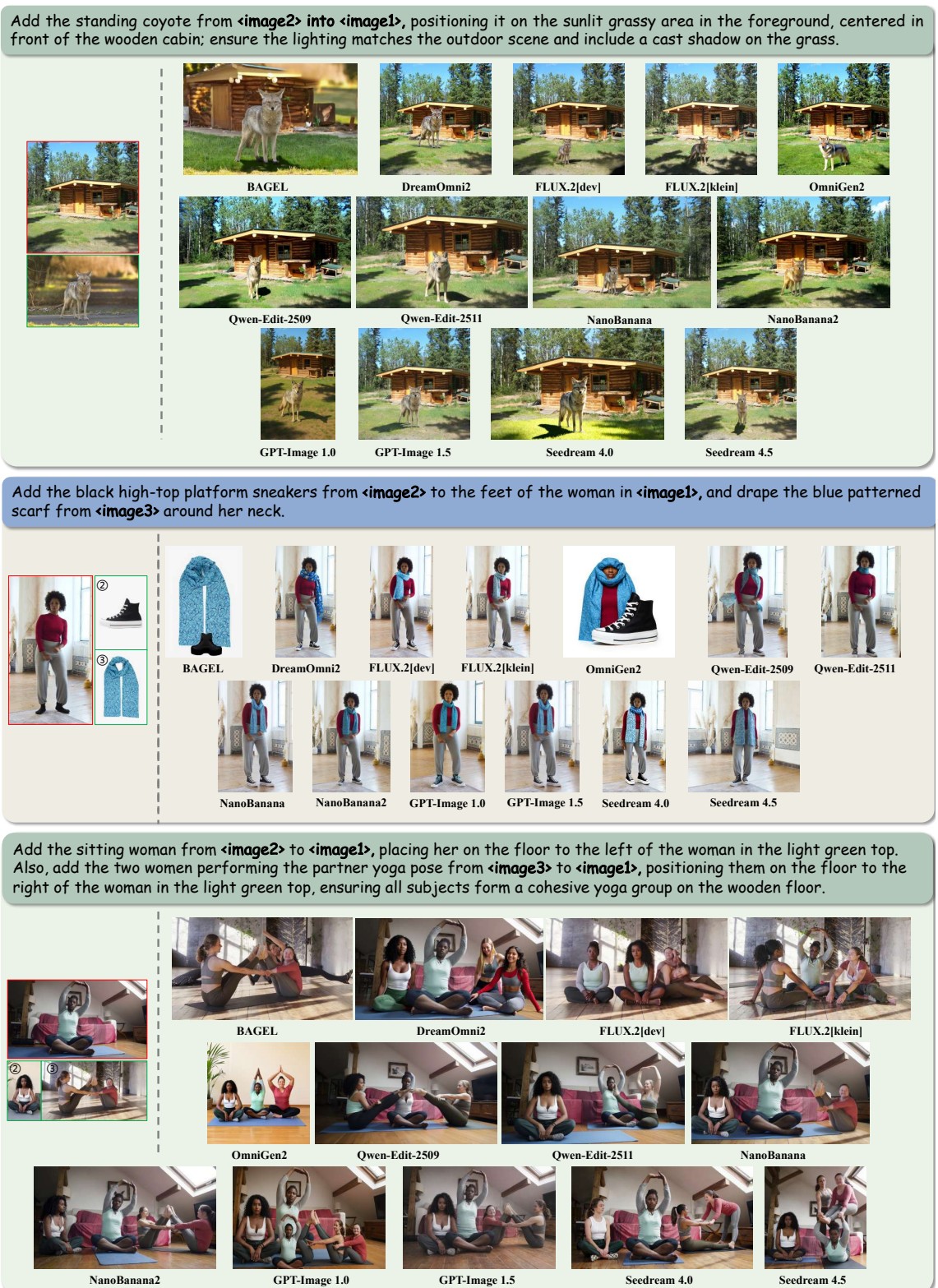

*Figure 18.* Qualitative Results on Multi-Reference Editing Task.

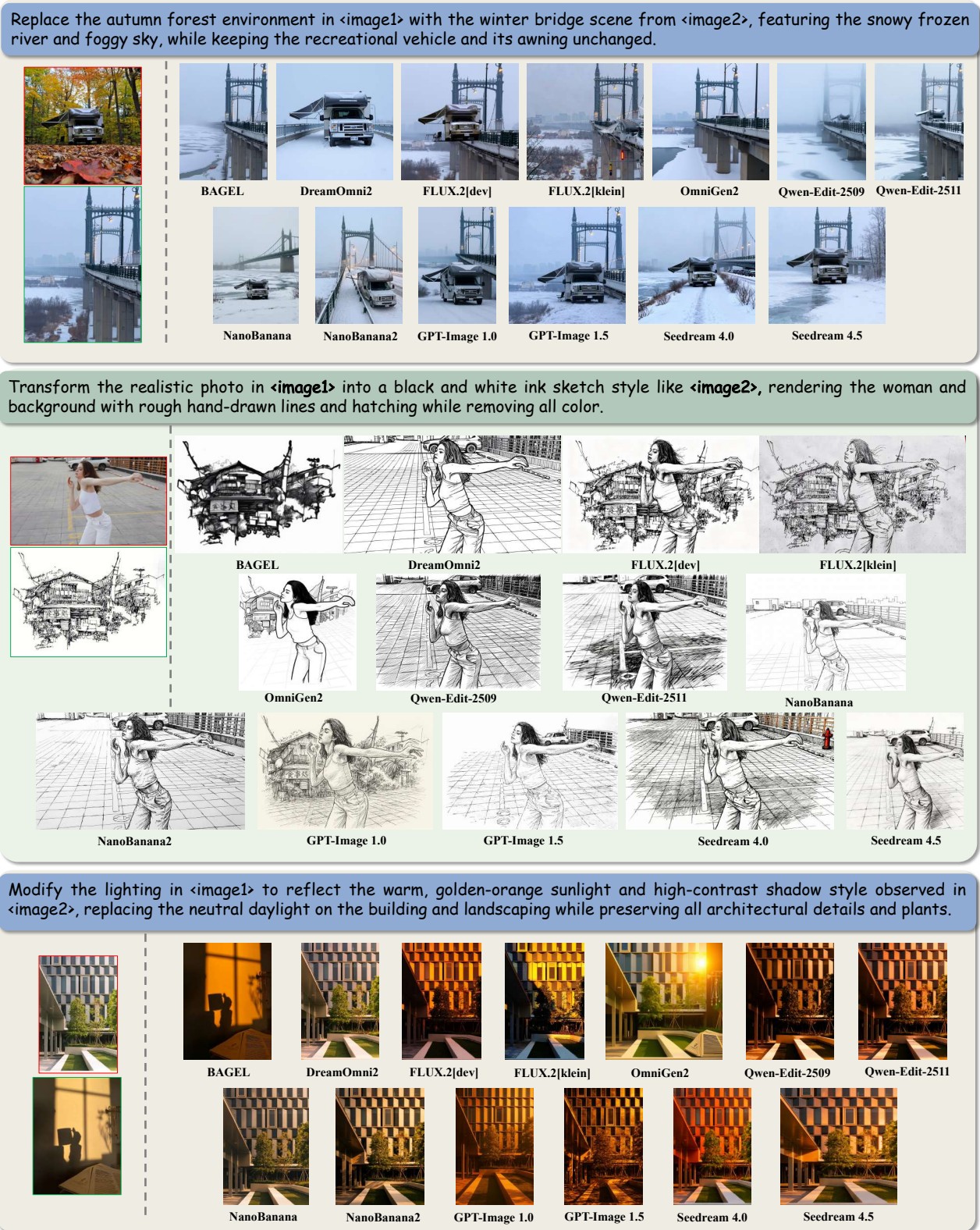

*Figure 19.* Qualitative Results on Multi-Reference Editing Task.

Move the cream-colored armchair indicated by **<image2>** to the left side of the room in **<image1>**, positioning it closer to the pink side table. Move the green patterned pillow indicated by **<image3>** to the floor, placing it on the rug in the bottom right corner of the image.

Remove the wavy mirror on the wall indicated by **<image2>**, the tall green plant indicated by **<image3>**, and the pink armchair indicated by **<image4>** from the living room scene in **<image1>**.

Replace the face of the man on the far left wearing a cream cardigan in **<image1>** with the face of the man with glasses from **<image4>**. Replace the face of the woman with dreadlocks in the center-left of **<image1>** with the face of the woman in the pink top from **<image2>**. Replace the face of the woman in the grey cardigan in the center-right of **<image1>** with the face of the woman with glasses from **<image3>**. Replace the face of the man taking the selfie on the far right of **<image1>** with the face of the man with curly hair from **<image5>**.

*Figure 20.* Qualitative Results on Multi-Reference Editing Task.

## D.4. Additional Analysis on Constraint Density in Multi-Image Creation

In the **Prompt Following** dimension, closed-source models consistently remain ahead and exhibit stronger robustness as the constraint density $K$ increases, whereas some open-source models show a clear performance drop once $K \geq 5$. This pattern indicates that when a prompt must satisfy multiple reference constraints simultaneously, models often prioritize a small subset of salient concepts and fail to fully align with all conditions.

In the **Concept Consistency** dimension, performance degradation with increasing $K$ is the most pronounced, with a sharp decline particularly from the Medium to Hard regime. This result suggests that **reference concept preservation** constitutes the central challenge in multi-reference creation. As the number of concepts increases, models become more prone to concept omission, incorrect concept placement, and intensified competition and interference among concepts, which prevents reliable retention of the defining attributes of each reference concept.

In the **Physical Realism** dimension, scores generally decrease as $K$ increases; however, some models exhibit a slight rebound from $K = 6$ to $K = 7$, following a trend similar to Prompt Following. We hypothesize that this rebound does not reflect a genuine improvement in physical modeling under high-density constraints. Instead, it is more plausibly explained by **goal simplification** under extreme complexity: models may implicitly abandon a subset of concepts and reduce cross-concept fusion conflicts, thereby producing images that appear more natural but are less complete. This behavior further highlights an inherent difficulty of creation tasks under **weak spatial layout constraints**, where models must autonomously decide how to fuse and arrange concepts, and consequently are more likely to deviate or degenerate into simplified generation as complexity increases.

Finally, **Image Structure** is the only metric that exhibits an increasing trend for most models. This is expected because the metric measures structural complexity and compositional organization. As the number of reference concepts grows, generated images typically contain more objects, attributes, and relations, which leads to higher structural complexity and therefore higher structure scores.

## D.5. Additional Qualitative Results

we present more qualitative comparisons of various models. Figure 15, Figure 17, and Figure 16 illustrate the multi-reference creation results. Figure 19, Figure 18, and Figure 20 illustrate the multi-reference editing results,

## D.6. Limitations and Future Work.

Although MICE-Bench systematically covers multi-reference image creation and editing, and characterizes model degradation under multi-concept composition via concept-wise partitioning and a constraint-density gradient, it still has several limitations. First, while our creation task provides stress-test settings with increasing constraint density, model behavior under larger-scale and more complex compositions (e.g., 12–15 reference inputs) remains unexplored. Second, our evaluation framework primarily relies on automated VLM-based judges across key dimensions, which may introduce evaluator bias and affect the stability and reproducibility of the conclusions. Third, the current task design focuses on static single-step generation and editing, and therefore does not fully capture the temporal interaction and multi-round editing patterns common in real-world workflows. As a result, it cannot comprehensively evaluate state maintenance, error accumulation, or cross-round consistency during continuous editing.

To address these limitations, we plan to extend MICE-Bench in three directions: (1) introducing larger-scale and more complex creation settings to strengthen stress testing for multi-reference generalization; (2) incorporating temporal and multi-round editing tasks to systematically evaluate stability, controllability, and consistency preservation over long editing horizons; and (3) adding a human-in-the-loop calibration mechanism that combines limited manual annotation with multi-evaluator aggregation to validate and calibrate automated judgments, thereby reducing systematic bias from a single evaluator and improving the reliability and interpretability of the benchmark.

