# OpenReview forum: "MICE-Bench: A Challenging and Comprehensive Benchmark for Multi-Reference Image Creation and Editing"
_ICML.cc/2026/Conference — ICML 2026 regular_

### Official Review · Reviewer_zkMr · 2026-03-12

**Soundness:** 3
**Presentation:** 3
**Significance:** 3
**Originality:** 2
**Overall Recommendation:** 4
**Confidence:** 4

**Summary:**

This paper introduces MICE-Bench, a comprehensive benchmark designed to evaluate Multi-reference Image Creation and Editing. Unlike most previous benchmarks limited to single-image conditioning, MICE-Bench focuses on heterogeneous concept composition across seven visual dimensions. It features a large-scale dataset of 3,119 high-quality test cases constructed through an automated pipeline with human-in-the-loop verification. By evaluating 13 state-of-the-art models using an 8-dimensional metric framework, the authors reveal that current models often rely on superficial composition rather than genuine synthesis, showing significant performance degradation as concept complexity increases. In general, this paper is well organized.

**Compliance With Llm Reviewing Policy:**

Affirmed.

**Final Justification:**

Most of my concerns have been addressed. Considering the strengths, weaknesses, and the rebuttal, I keep my original score.

**Key Questions For Authors:**

Please see the weaknesses.

**Limitations:**

yes

**Strengths And Weaknesses:**

Strengths:
+ The benchmark organizes tasks by constraint density (ranging from 2 to 7 concepts), allowing for a fine-grained analysis of how model performance evolves under increasing compositional pressure.

+ It bridges a critical gap in the field by spanning both creation and editing tasks across a broad unified concept space, including complex global transformations and localized modifications.

+ The authors developed a comprehensive 8-dimensional evaluation metric (covering physical realism, concept consistency, etc.) and validated it using multiple VLMs and supplementary metrics like FGCLIP to ensure robustness.

Weaknesses:
- Compared with recent existing benchmark (e.g., DreamOmni2), or the combination set of existing ones, what are the key differences of the proposed benchmark? For example, which new evaluation values does this method bring, compared with evaluation on ImgEdit-Bench, OmniContext, and DreamOmni2 together?

- Even though the benchmark becomes more comprehensive, the larger complexity may limit reproducibility or evaluation convenience.

- The paper compares datasets but does not extensively compare evaluation methodologies or demonstrate why the proposed metrics outperform existing ones.

---

> ### Author Rebuttal · Authors · 2026-03-31
>
> # Response to the comments of Reviewer zkMr
> We would like to thank you for taking the time to review our work. Your questions and comments are precious. We value each of them and have provided detailed responses below.
>
> > ### **Q1: Key Differences**
>
> We thank the reviewer for this question. The key difference is that MICE-Bench is not merely an accumulation of task coverage (like combining ImgEdit-Bench, OmniContext, and DreamOmni2), but a targeted diagnostic tool for a previously unanswered question.
>
> 1) **The Gap: Lack of Stress Testing for Multi-Concept.** ImgEdit-Bench focuses on single-image editing quality, OmniContext focuses on in-context generation under multi-image context, and DreamOmni2 focuses on multimodal instruction-based generation and editing. Even when combined, they do not systematically answer the fundamental question for multi-reference image generation models: how a model handles increasing concept density or growing structural complexity.
>
> 2) **The Solution: A Concept-Centric Unified Design.**
> MICE-Bench addresses this with a 7-category taxonomy and a controllable concept-density axis. By evaluating both task families under the same concept space and a systematically designed scoring framework, it enables fair, controlled comparisons across different difficulty levels.
>
> 3) **The Value: Diagnosing Failures Beyond Task Scores.** Rather than just reporting performance scores, an important contribution of MICE-Bench lies in analyzing specific failure patterns in multi-concept synthesis. At a basic level, it reveals which dimensions fail, such as concept consistency, physical realism, or editing consistency. At a deeper level, it further uncovers concept-specific degradation, cross-concept trade-offs, and phase-wise collapse as concept density increases. These findings help researchers pinpoint and fix specific weaknesses in multi-reference modeling.
>
>
> > ### **Q2: How to Ensure Reproducibility and Convenience**
>
> We thank the reviewer for this valuable feedback. While MICE-Bench provides a comprehensive evaluation, it is designed to be highly accessible, low-complexity, and easy to reproduce. Specifically:
>
> 1) **Fully Automated Pipeline.** The VLM scoring process is highly standardized. We will provide easy-to-use scripts to automate the entire pipeline, from input processing to final scoring based on our explicit criteria (Appendix Figs. 9–14). The entire scripts will be open-sourced after acceptance.
>
> 2) **Unified Format.** All 3,119 test cases in MICE-Bench will be released in a unified, cleanly structured JSON format alongside well-organized image directories, making it extremely easy for users to load and test their models.
>
> 3) **Removing Evaluator Barriers.** While our primary results rely on the powerful Gemini API, we understand that not all researchers have access to it. Therefore, we validated our benchmark using Qwen3-VL-8B (a smaller, open-source VLM). This ensures that future researchers can smoothly run our evaluation pipeline locally without relying on paid APIs.
>
> In summary, MICE-Bench successfully balances a comprehensive, multi-dimensional evaluation with low user complexity and high reproducibility.
>
>
> > ### **Q3: Why Proposed Metrics Outperform**
>
> We sincerely thank the reviewer for this insightful question. Our proposed metrics outperform existing evaluation paradigms for five main reasons:
>
> 1) **Comprehensive & Novel Dimensions.** Unlike previous works (e.g., DreamOmni2) which focus solely on instruction following, MICE-Bench uses 8 metrics for a 360-degree evaluation. We notably introduce novel multi-image metrics, evaluating multi-object concept consistency, physical realism, etc.
>
> 2) **Rigorous Scoring Criteria & Human Validation.** Each metric is governed by meticulously designed scoring guidelines and has undergone extensive human verification (see Appendix Figs 9-14) to maximize scoring accuracy and minimize bias.
>
> 3) **SOTA Evaluators.** We upgrade the evaluation pipeline by adopting the latest Gemini 3, UniPercept (for aesthetics/quality), and Qwen3-VL. This significantly improves evaluation accuracy compared to benchmarks using older versions (e.g., older Gemini 2.5 or Qwen2.5-VL).
>
> 4) **Open-Source Accessibility.** Unlike previous works (e.g., DreamOmni2) that rely solely on closed-source APIs, MICE-Bench offers an open-source evaluation pipeline based on Qwen3-VL. By providing both open and closed-source options, we lower the barrier for researchers and ensure the results are easily reproducible.
>
> 5) **Preventing Metric Hacking.** Our multi-dimensional approach prevents models from over-optimizing on a single metric (e.g., instruction following) while critically failing in others (e.g., physics or aesthetics), ensuring a reliable assessment of true capabilities.
>
> We will add a discussion in the revised manuscript to provide a thorough comparison between our proposed metrics and existing evaluation paradigms.

---

> > ### Author Rebuttal · Reviewer_zkMr · 2026-04-03
> >
> > Thanks for the rebuttal. Regarding Q2, an efficiency comparison (e.g., evaluation cost) between existing benchmarks is more convincing to verify its convenience, and the listed three points should be contained in most prior benchmarks or evaluation tools.
> >
> > As for Q3, the only advantage I can see from the response is that more metrics are introduced, but why are more metrics better than before? Does it align better with human evaluation? Except for this point, other claims like "SOTA Evaluators" and "Open-Source Accessibility" should be easily adapted to existing benchmarks, so they are not unique.

---

> > > ### Author Response · Authors · 2026-04-05
> > >
> > > > ### **Supplementary Response to Q2**
> > > |Benchmark|Support Task|Evaluator|API Cost|Deployment Cost|Benchmark Size|Total Cost|
> > > |-|-|-|-|-|-|-|
> > > |OmniContext|Multi-image Creation|GPT-4.1|`$0.008 / sample`|-|Creation: 400 cases|`$3.2`|
> > > |DreamOmni2|Multi-image Creation + Single-image Editing|Gemini 2.5|`$0.004 / sample`|-|Creation: 114 cases;  Editing: 205 cases (1 ref only)|`$1.3`|
> > > |MICE-Bench|Multi-image Creation + Multi-image Editing|Gemini 3|`$0.005 / sample`|-|Creation: 1872 cases;  Editing: 1247 cases|`$15`|
> > > |MICE-Bench|Multi-image Creation + Multi-image Editing|UniPercept & Qwen3-VL-8B|-|`$0.00002 / sample` ($1 per hour for one A100 GPU) |Creation: 1872 cases;  Editing: 1247 cases|`$0.06`|
> > >
> > > We thank the reviewer for this insightful question. The evaluation cost of a benchmark is determined by: **Total Cost = Per-Sample Cost × Benchmark Total Size.** We analyze each dimension separately:
> > >
> > > 1) **Per-Sample Cost**
> > >
> > > * **Similar API Cost.** Although MICE-Bench introduces multiple dimensions, our per-sample cost is similar to previous methods. This is because that a single API inference call can output all metric scores. Therefore, the main difference lies in the price per API call.
> > >
> > > * **Zero API Cost (Open-source Model).** While DreamOmni2 and OmniContext rely on paid commercial APIs, MICE-Bench also supports local deployment using open-source VLMs like Qwen3-VL-8B and UniPercept. This means users can evaluate with zero API cost, giving our benchmark a huge accessibility advantage.
> > >
> > > 2) **Benchmark Size.** Our benchmark size is indeed larger than previous ones, which naturally increases the total evaluation cost. However, this scale is strictly necessary. The larger size is due to comprehensive coverage of multi-reference scenarios and robust evaluation of multi-reference models, which smaller prior benchmarks fail to achieve.
> > >
> > > In summary, MICE-Bench does not increase the per-sample evaluation burden. The higher total API cost is simply because we provide a much more comprehensive and rigorous dataset. **More importantly, by offering the open-source local deployment option, we successfully reduce the total evaluation cost of this massive benchmark to near zero. Therefore, MICE-Bench achieves a perfect balance between rigorous, large-scale evaluation and high practical convenience.**
> > >
> > > > ### **Supplementary Response to Q3**
> > >
> > > We thank the reviewer for the insightful comment. While prior benchmarks (e.g., DreamOmni 2, OmniContext) focus narrowly on prompt following and reference consistency, human judgment is naturally holistic, considering factors like aesthetics and physical realism. Our 8 evaluation dimensions are specifically engineered to mirror these comprehensive human standards. To prove this, we conducted additional human evaluation experiments:
> > >
> > > **Comparison with Human Evaluation in Multi-Reference Creation.**
> > > |Model|DreamOmni2|OmniContext|MICE-Bench|Human Evaluation|
> > > |-|-|-|-|-|
> > > |Seedream 4.5|Evaluation has not been open-sourced|8.28|7.11|7.02|
> > > |NanoBanana 2|-|8.65|7.32|7.36|
> > > |GPT-Image 1.5|-|8.92|7.41|7.48|
> > > |DreamOmni2|-|7.24|5.64|5.52|
> > > |OmniGen2|-|7.45|5.85|5.65|
> > > |BAGEL|-|7.07|6.04|5.98|
> > >
> > > **Comparison with Human Evaluation in Multi-Reference Editing.**
> > > |Model|DreamOmni2|OmniContext|MICE-Bench|Human Evaluation|
> > > |---|---|---|---|---|
> > > |Seedream 4.5|Evaluation has not been open-sourced|Not supported|7.72|7.80|
> > > |NanoBanana 2|-|-|8.35|8.42|
> > > |GPT-Image 1.5|-|-|8.19|8.18|
> > > |DreamOmni2|-|-|6.78|6.65|
> > > |OmniGen2|-|-|4.88|4.79|
> > > |BAGEL|-|-|4.29|4.22|
> > >
> > > * **Human Evaluation Setup.** We selected 400 cases (300 creation, 100 editing) and invited 5 experts (each with 2+ years of experience in image generation/editing) to evaluate 6 models (1-10 points). Their scores were compared against MICE-Bench and OmniContext. (**Note: DreamOmni 2 is excluded due to its closed-source evaluation code; OmniContext does not support editing**).
> > >
> > > * **Experimental Results & Analysis.** MICE-Bench closely tracks human scores in absolute values. OmniContext exhibits severe “score inflation” across all models (e.g., scoring OmniGen2 at 7.45 vs. human 5.65). This occurs because OmniContext relies on narrow metrics: it rewards the mere presence of requested objects even if the image is physically distorted or aesthetically poor. In contrast, MICE-Bench penalizes these flaws through its comprehensive metrics, leading to an evaluation that much more accurately reflects human common sense.
> > >
> > > * **Visualization Results.** As shown in the visual examples **[Anonymous Link](https://anonymous.4open.science/r/ICML-Rebuttal-31CC/Rebbutal-Fig.pdf)**, while a model might satisfy the text prompt, the actual generated results are very poor. OmniContext assigns these cases high scores, whereas MICE-Bench correctly identifies these distortions and assigns lower scores, consistent with human judgment.
> > >
> > > We hope that our responses have addressed your valuable comments. **Please let us know if any points need further elaboration.**

---

### Official Review · Reviewer_rJTv · 2026-03-18

**Soundness:** 4
**Presentation:** 4
**Significance:** 3
**Originality:** 3
**Overall Recommendation:** 5
**Confidence:** 3

**Summary:**

This paper introduces a multi-reference benchmark for image creation and editing. The dataset spans up to seven heterogeneous visual concepts and systematically varies reference image constraint density (2–7), which enables evaluation under increasing compositional/editing pressure. The authors also benchmark 13 SOTA models and conduct extensive analysis of performance under varying constraint complexity.

**Compliance With Llm Reviewing Policy:**

Affirmed.

**Key Questions For Authors:**

1. Please clarify the fundamental differences between this work and DreamOmni2, beyond scaling in data size and constraint density.
2. Could the authors provide more insights into model behavior, specifically why certain models perform worse than others or struggle with particular concepts or higher constraint density?
3. Please also address the minor weakness noted above.

**Limitations:**

yes

**Strengths And Weaknesses:**

Strengths:
1. Well-motivated benchmark: the paper targets multi-reference compositional generation, as existing datasets are either not comprehensive or lack coverage of high concept density, contributing to its significance.
2. Technically sound: Curation and evaluation are well-explained, with comprehensive experiments. The use of multiple VLM evaluators (e.g., Qwen3, Gemini) improves reliability.
3. Well-executed: it evaluates performance across varying concept density (a novel aspect) and provides empirical insights into model behavior across different concept types/density.

Major weakness:
1. Incremental engineering effort: Both MICE and DreamOmni2 target multi-reference generation/editing. The paper needs to more clearly articulate what is fundamentally new beyond scaling up data size and concept density.
2. Overly empirical: the analysis reports performance trends but provides limited insight into why some models (Open-Source) fail on specific concepts or fail early at easy concept density.

Minor weakness:
1. Section 3.2 math notation $\lbrace c_j \rbrace_{j=1}^K \in \lbrace I_j \rbrace_{j=1}^K$ is a bit sloppy; better to explicitly describe one-to-one correspondence in words.
2. "Hybrid" editing subtask is used in section 3.2 and results tables but never formally defined in the main text.
3.  The more interesting analysis (concept specialization trade-offs, architectural bias findings) in Appendix C. is relegated to the appendix, while the main paper's insights focus on less novel observations like model evolution.

---

> ### Author Rebuttal · Authors · 2026-03-31
>
> # Response to the comments of Reviewer rJTv
> Thank you for the highly favorable evaluation. Your suggestions are highly constructive, and we will adopt them all to further improve the paper.
>
> > ### **Q1: Fundamental Difference**
>
> Thank you for the suggestion. We agree that this distinction should be stated more clearly. Beyond data scale and constraint density, the fundamental difference between MICE-Bench and DreamOmni2 lies in benchmark scope, design, and value.
>
> 1) **Benchmark Scope.**
> DreamOmni2 mainly evaluates whether a model can successfully complete multimodal generation/editing tasks. MICE-Bench expands the benchmark scope from basic task completion to multi-concept composition under increasing constraints, asking whether models can preserve, coordinate, and fuse heterogeneous concepts as reference complexity grows.
>
> 2) **Benchmark Design.**
> DreamOmni2 is primarily organized around task-level evaluation, whereas MICE-Bench is designed around a unified concept taxonomy, explicit constraint-density levels, and a multi-dimensional evaluation framework matched to this structure. As a result, tasks, references, and evaluation dimensions are jointly organized under a single concept-centric design.
>
> 3) **Benchmark Diagnostic Value.**
> DreamOmni2 mainly provides task-level evaluation, whereas MICE-Bench further supports diagnostic analysis. Multi-dimensional evaluation can expose gaps hidden by a single overall score. For example, although DreamOmni2 performs strongly under its own benchmark protocol, on MICE-Bench it shows much stronger physical realism (7.88) than concept consistency (4.75), meaning realistic-looking outputs can still fail to preserve the referenced concepts. Beyond this, MICE-Bench also supports deeper analyses such as concept-specific degradation, cross-concept trade-offs, and phase-wise failure under increasing constraint density.
>
> > ### **Q2: Why Certain Models Struggle with Higher Constraint Density.**
> >
>
> **Rebuttal Table 1: Concept-Level Retention Comparison between DreamOmni2 and GPT-Image-1.5. Avg. Retention is averaged over all constraint density levels from K=2 to K=7. Total Decay measures the retention drop from K=2 to K=7. Peak Drop reports the interval with the largest single-step decline and its magnitude.**
>
> |Concept|DreamOmni2 Avg. Retention|GPT-Image-1.5 Avg. Retention|DreamOmni2 Total Decay (K=2→7)|GPT-Image-1.5 Total Decay (K=2→7)|DreamOmni2 Peak Drop|GPT-Image-1.5 Peak Drop|
> |:---|:---|:---|:---|:---|:-----|:--------|
> |Identity|55.01|94.98|-46.49|-16.67|K=4→5(-25.33)|K=6→7(-16.67)|
> |Appearance|61.47|88.45|-38.42|-13.34|K=5→6(-21.61)|K=3→4(-6.99)|
> |Material|41|93.55|-71.41|-22.22|K=2→3(-22.00)|K=6→7(-18.77)|
> |Object|39.08|88.86|-70|-21.05|K=4→5(-30.33)|K=3→4(-9.34)|
> |Scene|28.13|95.51|-58.07|-7.66|K=4→5(-24.73)|K=6→7(-7.54)|
> |Style|21.42|74.2|-46.46|-24.6|K=2→3(-27.72)|K=4→5(-9.78)|
> |Control|31.65|72.68|-30.35|-28.89|K=2→3(-26.53)|K=5→6(-27.44)|
>
> We agree that a deeper analysis of why certain models struggle under higher constraint density would strengthen the paper. To illustrate this, we compare a weaker model (DreamOmni2) with a stronger commercial model (GPT-Image-1.5) in Rebuttal Table 1. The comparison suggests that weaker models face severe bottlenecks in three areas:
>
> 1) **Concept Representation.** DreamOmni2 achieves an overall concept retention of only 39.68%, far below GPT-Image-1.5’s 86.89%, indicating weaker encoding and preservation of reference features.
>
> 2) **Cross-Concept Fusion.** As density increases, DreamOmni2 decays much more sharply (-51.60% vs. -19.20%), with especially large drops in Material (-71.41%) and Object (-70.00%), suggesting stronger concept interference.
>
> 3) **Global Coordination.** For global concepts like Style and Control, weaker models collapse earlier. DreamOmni2 sees its sharpest drop at K=2→3 (-27.72%), while GPT delays this collapse until K=4→5 or K=5→6.
>
> According to public technical reports for models such as DreamOmni2 ,OmniGen2, and Qwen-Image-Edit, these differences may reflect not only overall model strength, but also differences in training objectives and data construction associated with different model capabilities. Their performance differences on MICE-Bench likely reflect not only model strength, but also whether their training data and tasks sufficiently cover high-density, multi-reference, heterogeneous concept composition.
>
> > ### **Q3 (Minor): Clarify Mathematical Notation & Define the "Hybrid Editing Task".**
>
> We thank the reviewer for these suggestions and will incorporate them in the revision:
>
> 1) **Mathematical formulation.** In Sec. 3.2, we will refine the notation and explicitly clarify the one-to-one mapping between references and concepts.
>
> 2) **Task definition.** We will add a formal definition of the “Hybrid Editing” subtask in the main paper.
>
> 3) **Content organization.** We will move key analytical discussion from Appendix C to the main paper to strengthen the experimental explanation.

---

> > ### Author Rebuttal · Reviewer_rJTv · 2026-04-01
> >
> > The authors have fully addressed my concerns.
> > (a) They clearly clarify the differences from DreamOmni2 beyond dataset scaling.
> > (b) They provide additional analysis explaining 2 open-source model behaviors under different concept densities.
> > (c) The minor issues (notation, definition etc.) have been resolved.

---

> > > ### Author Response · Authors · 2026-04-01
> > >
> > > We are glad that our responses have fully addressed your concerns.
> > >
> > > Thank you for your time and constructive suggestions, which have significantly strengthened our paper.

---

### Official Review · Reviewer_ADmG · 2026-03-22

**Soundness:** 3
**Presentation:** 3
**Significance:** 2
**Originality:** 2
**Overall Recommendation:** 3
**Confidence:** 2

**Summary:**

The paper proposes MICE-Bench, a large multi-reference image creation and editing benchmark to evaluate image editing and creation qualities. The dataset is the largest-scale compared to existing benchmarks, and the data curation process underwent both automatic and manual filtering and checking. The paper also proposes multiple evaluation metrics, measuring image generation and editing tasks both in fildelity to prompts, reference images, and physical realism. The paper also conducted experiments running both closed source and open source image generation models on the benchmark and reported the rankings and analysis. Overall, the paper looks like an effortful submission with many contents.

**Compliance With Llm Reviewing Policy:**

Affirmed.

**Key Questions For Authors:**

1. Comparison with existing benchmarks. Why is your benchmark better at evaluating image generative models? I think this is crucial and missing from the current submission. (see my comment under significance)
2. In general, the authors did not explain why we need to increase the number of reference images and evaluation metrics. Does the current model already saturate on the current evaluation benchmarks with 2-4 reference images? For each evaluation metric, how do you make sure that they are evaluating (roughly) independent axes of the generated quality? Otherwise, I would prefer keeping the number of metrics small for easier understanding.
3. How do you make sure the metrics proposed are consistent across multiple runs? It seems that some of the metrics uses continuous scores; different runs of VLM could result in different scales of scores.

Addressing the above questions would make me more convinced that this benchmark is a valuable contribution to the field.

**Limitations:**

yes.

**Strengths And Weaknesses:**

Soundness: The proposed benchmark's data curation and evaluation details are outlined in the paper and the supplementary. They all seem sound and well executed.

Presentation: The presentation quality is OK. The annotations shown in Figure 4 is confusing. Why there's crosses, checkmarks, and boxes? The authors should provide more detailed explanation about what's going on in the caption. Also, the prompt used to for generation should be provided, as the reference images alone is ambiguous and it's not quite clear why the models have failed to achieve the desired generation result.

Significance: The problem that the paper addresses is important. However, there are many relevant works in this area, all of them proposing their own benchmarks. I found it hard to justify why we need another benchmark. I understand that the paper has proposed a large-scale benchmark with more reference images, but I think some kind of a comparison with past benchmarks is necessary to justify why the proposed benchmark is better. For example, compared to some of the benchmarks listed in related works (Tab.1), does this capture some missing pit-fall or insight within current image generation models that other benchmarks do not capture? Why do we need the extra dimentions of evaluation compared to the previous benchmarks' protocol? I would like to see some analysis and justification on this front -- Otherwise I don't see why more metrics + benchmark size is better for a benchmark.

Originality: Not so much. The paper proposes a benchmark with new evaluation metrics judging different front of the generated images w.r.t. image and editing. However, most of the metrics are prompting LLMs with different prompts, which is fairly standard for evaluation, and the task that the paper considers already include many similar benchmarks, just in smaller scale.

---

> ### Author Rebuttal · Authors · 2026-03-30
>
> # Response to the comments of Reviewer ADmG
> We would like to express our most sincere gratitude for your time and effort. We value each of your suggestions and provide the following responses.
>
> > ###  **Q1: Advancements Over Existing Benchmarks**
>
> We emphasize that MICE-Bench is not simply a larger version of existing benchmarks. Instead, it differs from prior work in three important ways：
>
> 1) **Why a New Benchmark is Needed.** Existing benchmarks mainly test basic multi-reference usability, often without a concept-centric formulation, making fine-grained failure diagnosis difficult. This leaves a more general benchmark gap: whether models can robustly represent, resolve, and fuse heterogeneous visual concepts as constraints increase. MICE-Bench is designed to address this gap.
>
> 2) **Concept-Centric Benchmark Design.** Based on the above viewpoints, MICE-Bench is built on the principle of Heterogeneous Concept Composition. Data construction and evaluation pipelines are also entirely concept-driven.
>
> 3) **In-depth Diagnostic Insights.** Beyond performance rankings, MICE-Bench offers deep insights (concept-specific degradation, trade-offs across concepts, phase-wise failure patterns, etc). These findings provide a clear roadmap for improving model robustness in multi-reference scenarios.
>
> > ###  **Q2-1: Why We Need More References**
>
> The number of reference images is critical for multi-reference tasks. If a benchmark only tests 2-4 images (**which often belong to the same concept in DreamOmni2**), it merely proves that a model has basic multi-image capabilities. It cannot answer more critical questions: How does the model perform as concepts become more diverse? Where is the breaking point, and is failure caused by interference between concepts? Specifically, we increase reference images for two reasons:
>
> 1) **Modeling Concept Constraint Density.** We do not increase the number of images simply to “have more images.” In MICE-Bench, each reference image introduces a distinct visual concept (Sec 3.1), increasing the image count is essential to analyze “concept constraint density.”
>
> 2) **Aligning with Model Trends.** Modern models support more concepts (e.g., Seedream 4.0/4.5 takes 10+ references). Benchmarks must evolve beyond 2-4 images setup to reflect real-world applications.
>
> > ###  **Q2-2: Why Diversify Metrics & How to Ensure Metric Independence**
>
> Our goal is to align each metric with a specific quality dimension and failure mode, forming a complementary evaluation framework. This is necessary because an overall task score can hide imbalanced behavior. This design also reduces “metric hacking” via single-metric overfitting. For example, although DreamOmni2 performs strongly on its own benchmark with an overall score, on MICE-Bench its physical realism (7.88)  is much higher than its concept consistency (4.75). We ensure the independence of metrics through two strategies:
>
> 1) **Distinct Evaluation Goals.** Each metric targets a unique aspect. For instance, prompt following assesses alignment with text conditions, concept consistency measures how well the reference subject is preserved, and physical realism identifies logical errors.
>
> 2) **Independent Scoring Criteria.** As shown in Appendix Figs 9–14, each metric features an independent, fine-grained scoring criteria. These criteria are strictly separated to avoid overlap.
>
> > ###  **Q3: How to Ensure Metric Consistency Across Multiple Runs**
>
> Using VLMs for evaluation is common in image generation and editing tasks (e.g., ImgEdit and ICE-bench). Although VLMs naturally have some randomness, we design minimizes this variation to ensure our conclusions are reliable via three ways:
>
> 1) **Strict Scoring.** Detailed step-by-step criteria (Appendix Figs. 9-14) make VLM judgments more stable and consistent.
>
> 2) **Deterministic Configurations.** We fix API parameters for Gemini and keep deployment and inference settings consistent for open-source evaluators such as Qwen3-VL.
>
> 3) **Cross-Validation.** We validate the main findings with multiple evaluators, including Qwen3-VL (App. D.2) and FGCLIP (App. D.3), and observe consistent rankings and trends.
>
> > ###  **Q4 (Minor): Regarding the confusion about Fig.4**
>
> We will update the manuscript to clarify the symbols in Fig.4:
>
> - **Checkmarks (✓):** successfully generated or preserved concepts.
> - **Crosses (✗):** missing, distorted, or instruction-inconsistent concepts.
> - **Boxes:** the corresponding local regions.
>
>
> Prompts in Fig.4:
>
> * **Prompt 1:** A realistic full-body photo of the woman from [image2] on the road from [image5], mimicking the pose of [image4]. She wears the hat from [image1] and holds the roses from [image3] against a natural background.
>
> * **Prompt 2:** Add the fedora from [image2] to the man in the beige shirt, the necklace from [image3] to the woman in the grey shirt, and the wristwatch from [image4] to the man in the maroon shirt, all in [image1].

---

> > ### Author Rebuttal · Reviewer_ADmG · 2026-04-03
> >
> > Listed in comments

---

> > > ### Author Response · Authors · 2026-04-03
> > >
> > > Thank you again for reading our rebuttal and for the follow-up.
> > >
> > > We noticed your status is marked as "Partially resolved or unresolved" with the reason "Listed in comments."
> > >
> > > In our rebuttal, we deeply valued your feedback and aimed to comprehensively address all your core questions:
> > >
> > > * **Difference from existing benchmarks.** MICE-Bench is designed to address a previously underexplored diagnostic question: not just whether a model can complete a multi-reference task, but where and why it fails as heterogeneous constraints increase. Its core contribution is a unified concept taxonomy, controlled constraint-density design, and a matched evaluation protocol that together enable attributable analysis.
> > >
> > > * **Why more references and multi-dimensional evaluation are necessary.** More references and multi-dimensional evaluation are necessary because we study failure modes under increasing constraint density, not just basic multi-image usability. This is why it matters that models no longer saturate at 2-4 references. A single overall score cannot reveal which constraints fail first, nor whether failure comes from weak concept preservation, weak realism, or weak editing consistency.
> > >
> > > * **Why the metrics are independent.** Our metrics are designed to be complementary rather than redundant, with distinct evaluation goals and separated scoring criteria. This is what allows different failure modes to be diagnosed instead of being conflated under a single score.
> > >
> > > * **Why the metrics are consistent across runs.** We reduce run-to-run variation through fixed scoring rubrics, consistent evaluation settings, and controlled VLM configurations, and we further validate the conclusions through cross-evaluator comparisons, including Gemini 3 as the main judge, Qwen3-VL-8B as an open-weights evaluator, and FGCLIP as an additional alignment metric. We will make this diagnostic role and evaluation design much more explicit in the revision.
> > >
> > > **To help us move forward, could you please clarify which of your original comments need further explanation?**
> > >
> > > **We are eager to provide more in-depth answers to fully resolve your concerns.**

---

### Decision · Program_Chairs · 2026-04-30

**Decision:**

Accept (regular)

**Comment:**

This paper proposes a benchmark for image creation and editing tasks involving multiple reference images. The main concerns from the reviewers included novelty (how the benchmark differentiates itself from or complements existing ones), the quality of the proposed benchmark such as its diversity (e.g., zkMr), and the reliability of the VLM metrics (e.g., ADmG and zkMr). Most of these concerns were addressed by the rebuttal. The remaining concerns, such as quantifying the diversity of the benchmark images and the variance of the metrics, do not undermine the core contribution of the benchmark, which is curated to capture the core challenge of multi-reference image generation and editing models. I recommend the authors strongly consider the reviewers’ recommendations to sharpen the argument and positioning of the paper, and ensure all promised modifications are included in the final version.